# Navigating the leaky pipeline: Do stereotypes about parents predict career outcomes in academia?

**Vasilena Stefanova** [ID]*, **Ioana Latu**

School of Psychology, Queen's University, Belfast, United Kingdom

* vstefanova01@qub.ac.uk

## Abstract

The motherhood penalty seemingly reflects a preference to hire female professionals who are not parents compared to mothers, however, little is known about whether this effect is attributable to parent stereotypes per se. Study 1 assessed the content of the parent-academia stereotypes of 180 individuals working in Education and revealed stronger stereotypical associations of fathers with academia than mothers. Study 2 investigated what parent-academia stereotypes predict in terms of endorsements for hiring men versus women in a mock hiring task set in an academic context. Academics (N = 112) evaluated mock job candidates for an Assistant Professor post while the gender, parental status and leave status of the candidates were manipulated. The findings showed that parents were significantly less likely to be endorsed to be hired than non-parents, regardless of gender. Parent-academia stereotypes led to biased hiring recommendations, such that a greater endorsement of parent-academia stereotypes predicted a reduced likelihood to endorse hiring parents compared to non-parents. Implications for reducing parent stereotypes in academic contexts are discussed.

## Introduction

Although significant progress toward gender equality has been made in academia, women continue to face challenges associated with professional advancement. This effect is reflected in the consistent underrepresentation of female academics in senior positions across academic fields, thus giving rise to the leaky pipeline phenomenon [1–3], which shows attrition of female academics at each critical transition of the career ladder [4]. A gender imbalance is visible in women's academic eminence [5], their visibility in academic science, and the opportunity to establish role models for junior female academics [6–8]. These imbalances may point to underlying gender biases which hinder the advancement and promotion of women in academia [9–11]. Failure to recruit and retain female faculty through academic promotion leads to gender imbalances in university leadership [3]. The lack of women in senior academic positions shows a severe underrepresentation of women in academic decision making, as members of committees and recruitment panels. This gender imbalance in scientific leadership results in

**Data Availability Statement:** All data files are available from the Open Science Framework (OSF) database (https://osf.io/87vyq/?view_only=54629e03c435483182e80f82f00260c7).

**Funding:** The authors received no specific funding for this work.

**Competing interests:** The authors have declared that no competing interests exist.

an underuse of the expertise and skills of a significant part of the Higher Education workforce [12]. It is therefore essential to address gender inequalities in academia, in order to ensure a diversity of perspectives in scientific leadership.

Using the theoretical framework of role congruity theory [8] and experimental methods employing mock hiring tasks that are widely used in the field [13, 14], the present paper focuses on examining these gender biases in academia, especially as they interact with biases about parents. Investigating the role of gender and parent biases in academia as a career path is especially important, as the initial stages of an academic career involve a significant delay in financial and organisational rewards and often coincide with a period when many people are building a family. Academics are "in school" until later in their lives compared to other professionals and often rely on temporary contracts that are renewed every few years until they secure a permanent position, while continuously having to demonstrate high productivity and commitment. Thus, advancing through the academic ranks could create significant challenges for early career researchers who are raising families, as they would often need to make career decisions while their children are young [10, 15].

## Gender biases and parent biases

Gender biases were previously documented in academic hiring decisions, where both male and female evaluators showed a preference to hire male job candidates compared to female candidates with an identical record. They also rated the male candidates higher in terms of experience and competence [16]. Faculty were also found to show gender biases when hiring students for the post of laboratory manager, such that male students were evaluated more favourably and were offered a higher starting salary compared to female students with identical application materials [17]. Lower perceived competence judgments on the part of academic staff, fewer rewards, and less mentoring received by junior academics may increase the likelihood that women leave academia, contributing to the leaky pipeline phenomenon.

Another major challenge that female academics face involves combining academic aspirations with family and caregiving responsibilities. Balancing between a scientific career and parenting creates tension for female academics in faculty positions and research roles who are faced with pressures to secure funding for research projects, submit publications, and meet their teaching and research commitments [10, 18]. Williams and Segal reported that women who have children soon after receiving their PhDs are significantly less likely to achieve tenure than men who have children at the same point in their academic career [19]. It has therefore been suggested that mothers experience heightened barriers in the workplace that go beyond those associated with gender per se [20]. Although the contribution of fathers to childcare has significantly increased in recent decades, mothers still perform more caregiving duties [21], thus the ability of female faculty members to perform academic work may be more significantly impacted by caregiving responsibilities compared to male faculty members [22, 23]. This tension created by the necessity to balance dual responsibilities may further add to the leaky pipeline effect and could reinforce stereotypes about women which put them in restrictive roles and hinder their progress.

The maternal wall and motherhood penalty are terms that refer to the discrimination that mothers in employment or those seeking employment face [13, 19]. In a study by Correll, Benard and Paik which was conducted in the US in 2007, mothers applying for a marketing position were significantly less likely to be recommended for hire and were offered a significantly lower starting salary than equally qualified childless women [13]. This effect was mediated by evaluators' reported beliefs that mothers were less competent and less committed to their jobs compared to childless women. In contrast, a fatherhood premium was found for

men, where evaluators gave more favourable ratings to fathers compared to equally qualified childless men. These effects were supported by a study by Cuddy, Fiske and Glick, conducted in the US in 2004, which showed that consultants who were mothers were significantly less likely to be hired and recommended for promotion or job-related training compared to childless women [14]. These findings highlight differences in the way women and men are perceived in the context of parenting and career development, which could lead to differential professional outcomes. However, these studies were conducted within an industry job context. Quantitative, experimental evidence of the motherhood penalty in an academic context is currently lacking in the field. Our study aims to address this gap by focusing on academia as a work environment. Investigating academic contexts is especially important given that they pose unique challenges. Importantly, academic promotion and tenure criteria are traditionally based on a male-gendered career trajectory which could potentially create further tension for early-career female academics who wish to raise families [10].

## The role of parental leave

Generally, as there are more mothers taking parental leave than fathers, most research on the parent biases in organisational contexts has focused on mothers [23, 24]. However, in some cases the discriminatory effects may also extend to men in caregiver roles [25, 26]. Judiesch and Lyness reported that employees who took parental leave had a lower likelihood of promotion compared with employees who did not take leave, suggesting that leave-taking could diminish the fatherhood premium effect [27]. Leave status was found to impact evaluators' perceptions particularly of male employees, such that men who took paternity leave received lower ratings for workplace altruism and compliance compared to men who did not take leave [26]. Thus, it has been suggested that men who choose to take paternity leave or opt for more flexible work schedules due to caregiving responsibilities may also be penalised in the workplace [14, 28]. This effect is not consistent with the fatherhood premium, suggesting that more research is needed to investigate the interactions between gender, parental status and leave status, as well as their impacts on career outcomes.

Overall, the aim of the present studies was to assess if there was evidence that parent-academia stereotypes predict biased hiring decisions, reflected in a reduced likelihood to recommend hiring a parent compared to a non-parent for an academic post. We approached stereotypes from a social cognitive perspective, according to which stereotypes are associations that reflect the beliefs people have about corresponding social groups [29]. Additionally, we aimed to assess whether the job candidates' gender, parental status, and leave status impact hiring recommendations in academia independently and interactively.

## Theoretical development and predictions

A theory that has been proposed to explain the motherhood penalty is the role congruity theory [8], which indicates that discrimination stems from the discrepancy between individuals' perception of members of a social group and the stereotypical requirements of members of that social group. In the context of parenting and career progression, discrepancies often arise between the gendered norms of the ideal worker and the ideal mother. In an organisational context, employees are expected to be ideal workers, which involves being highly dedicated to one's job and being always available to perform work duties. A woman's motherhood status would be incongruent with the image of the ideal worker, which in turn would lead to motherhood being stereotyped negatively when evaluating a worker's job performance [30]. Thus, as an individual's caregiving role becomes salient, they may be perceived as less fitting to the job–and consequently, may be less likely to be hired. In contrast, working mothers who choose not

to take maternity leave after having a child might be perceived as not fitting into the norm of the ideal mother and being less committed to their children, and might therefore be also judged more harshly [31]. This tension created by incongruent societal expectations for mother and worker roles could underlie the motherhood penalty in career progression. In the current work, we propose that this tendency might be especially likely for those evaluators who hold mother-career stereotypes, such that they are less likely to associate mothers with academic careers. Such strong stereotypes mean that those individuals perceive the two categories ("motherhood" and "academic career") as being more incongruent, thus leading to a greater perceived discrepancy between the two roles (mother and academic), which could in turn lead to more negative evaluations of academics who are also mothers, consistent with the role congruity theory.

A model that could further add to our understanding of the motherhood penalty and fatherhood premium in career outcomes is the stereotype content model, which evaluates stereotyped groups in relation to warmth and competence [32]. Mothers are typically stereotyped as being high in warmth and low in competence which may evoke responses involving expressions of pity and patronising behaviours, which may in turn lead to discrimination [14]. A traditional cultural belief in Western society attributes the role of primary caregiver to women and the role of primary breadwinner to men [33]. This caregiver/breadwinner distinction gives rise to cultural stereotypes that result in working mothers being viewed as more nurturing than professional and therefore less committed to work and likely to put less effort and time in their job after having children [14]. Thus, consistent with the stereotype content model according to which the caregiver stereotype implies high warmth and low competence, mothers who apply for high-powered jobs may face biased evaluations, resulting in the motherhood penalty [14]. In contrast, the breadwinner stereotype emphasises high competence and low warmth and is traditionally associated with "masculine" traits such as assertiveness and dominance, which connect to leadership and may underlie the fatherhood premium in career progression. However, according to the role congruity theory, men who do not fit into the breadwinner stereotype schema might also be penalised in the workplace. This could potentially be applied to fathers who take paternity leave, who might be perceived as primary caregivers and might therefore receive less favourable evaluations due to the clash between cultural stereotypes about masculinity and caregiving [26].

We additionally hypothesise that these biased evaluations of mothers and fathers in academia may be attenuated or increased by individual differences, particularly evaluators' own stereotypes about parents and academic careers. More generally, the degree to which members of a social group are perceived as being congruent or incongruent with certain societal roles may depend on the extent to which evaluators stereotype those social groups. For example, gender stereotypes were found to underlie the gender salary gap, such that individuals who strongly endorsed gender stereotypes were more likely to allocate a higher salary to male employees compared to female employees [34]. Additionally, other individual differences such as attitudes toward women in leadership positions were found to mediate the link between role congruity and gender bias in leader evaluations [35]. These previous findings suggest that individuals' own level of stereotypical associations may impact their behaviours toward individuals from a particular social group.

In the context of parent-academia stereotypes, individuals who perceive a mismatch between career prototypes and parent stereotypes could experience more negative attitudes towards mothers in academic positions and this could predict biased evaluations of job candidates who are mothers in a hiring context. As such, we hypothesise that academics' endorsement of parent-academia stereotypes will impact their hiring decisions in an academic context, such that academics with stronger parent-academia stereotypes (i.e., whose

stereotypical associations show a stronger incongruity between academia and motherhood) would be less likely to recommend hiring a job candidate who is a mother compared to a job candidate who is a non-parent. After investigating the content of parent-academia stereotypes in Study 1, we proceed to test the above predictions using an experimental method involving a mock hiring decision, as is commonly used in the previous literature [13, 14]. Importantly, Study 2 employs a sample of actual academics.

At present we do not fully understand what drives the gender gap in academic leadership. As such, an investigation into the impact of parent-academia stereotypes on career outcomes could yield not only valuable theoretical insights in a career development context but could also aid in the creation and application of interventions that aim to reduce these stereotypes in academic contexts.

## Study 1

In order to examine the nature and expression of parent-academia stereotypes, we conducted an initial exploratory study which assessed stereotypes specific to parents, gender and academia in a sample of individuals working in Education. The main aim of Study 1 was to investigate whether individuals working in Education associate mothers more with family than fathers and fathers more with academia than mothers, in order to gain insight into parent-academia stereotypes–a topic which, to the best of our knowledge, has not been investigated before. We also aimed to investigate to what extent participant demographic characteristics, such as gender, parental status and occupation, predict the strength of parent-academia stereotypes.

### Methods–Study 1

**Participants–Study 1.** One hundred and eighty participants working in the field of Education took part in the study via Prolific, an online crowdsourcing platform for participant recruitment, and received £3 for their participation as per Prolific's recommendation. We recruited a sample of individuals working in Education in Study 1 to explore parent-academia stereotypes more broadly. Academia is a part of Education and we aimed to probe parent-academia stereotypes in a broader population, encompassing the field of Education in general. All participants were above 18 years of age, ranging from 18–66 years ($M_{Age}$ = 38 years; $SD_{Age}$ = 11.47). All participants were from the UK, were native speakers of English, and identified themselves as either male (52) or female (128). Seventy-two participants reported being parents (including step-parents and adoptive parents), and 108 reported not being parents. Seventy-six of the respondents reported being professionally involved in academic research or research and teaching combined, while 104 had non-academic jobs, such as college and university administration, IT support, secondary school or college teaching, student unions and campaigns, disability support, etc.

**Measures and procedure–Study 1.** The current study was approved by the Faculty of Engineering and Physical Sciences Research Ethics Committee at Queen's University Belfast (reference number EPS 18_178). The survey was implemented on Qualtrics, a web-based tool for survey implementation and online data collection. After reading the informed consent and agreeing to take part in the study, all participants completed the parent-academia stereotypes task and then they provided demographic information. Finally, participants were debriefed and thanked for their participation.

*Parent-academia stereotypes task.* The survey assessed associations between "mother"/ "father" and "academia"/ "family" involving 48 questions, of which 24 assessed associations for 'mother' and 24 assessed associations for "father" with words from the "academia" category

(professor, research, faculty, lecturer, publication, promotion, scholarly, salary, tenure, eminence, grant, conference, $\alpha$ = .96) and the "family" category (children, kinship, marriage, nurture, infant, household, relatives, housework, upbringing, baby, home, caregiving, $\alpha$ = .92). The "academia" and "family" category stimuli were pre-tested to ensure that they clearly belong in only one of the two categories. Eighteen academics took part in the pre-test and completed a questionnaire assessing the extent to which the words from the lists fit into the "academia" category and the "family" category. The questions were phrased as "To what extent does the word "professor" belong in the "academia" category?" and responses were given on the 7-point scale. The stimuli that were chosen to be included in the academia word list above were rated as belonging to the academia category with a mean score of 6 out of 7 or higher at the pre-test and to the family category with a mean of 2 or lower. Likewise, the stimuli chosen to be included in the family word list were rated as belonging to the family category with a mean score of 6 out of 7 or higher at the pre-test and to the academia category with a mean of 2 or lower.

In measuring stereotypes, in order to avoid social desirability bias, the questions measured participants' cultural stereotypes about parents as they relate to academia and family, and phrased as: "To what extent do you think people in our society associate the word "mother" with the word "professor"?" Responses were given on the 7-point Linkert scale where 1 was "not at all" and 7 was "extremely". This decision is also supported by evidence of overlap between personal beliefs and cultural stereotypes of other groups [36] and consistent with the stereotype content model which assesses stereotypes from a shared cultural perspective [32]. The social projection model implies that social perceivers tend to estimate prevalent cultural stereotypes based on their own personal beliefs, in that they tend to assume their own beliefs are more common in the general population [37]. Thus, projection results in a positive correlation between a person's beliefs about the characteristics of a particular social group and their perceptions of the cultural stereotype about the social group [36]. Therefore, in research we consider perceptions of cultural beliefs to proxy personal beliefs.

The order of presentation of father/mother question clusters was counterbalanced across participants, whereby half of them responded to questions about associations with "father" first and "mother" after and half to questions about "mother" first and "father" after.

*Demographics*. The demographics questionnaire involved questions about participants' gender, age, country of origin, native language, parental status, ethnicity, level of education, area of Education they work in and for how long they have worked there.

**Power analysis–Study 1.** A post-hoc statistical analysis was performed on G*Power for a repeated measures ANOVA with between-subjects and within-subjects factors, a sample of 180 and thirty-two groups with an effect size of $f(U)$ = 0.55, calculated based on $\eta2p$ = 0.23 as in SPSS, alpha = 0.05. The analysis showed that the current test had power = 0.83.

## Results and discussion–Study 1

**Analytic plan.** Our first goal was to investigate the content of parent-academia stereotypes by assessing the extent to which mothers are more strongly associated with family than academia compared to fathers and fathers more with academia than family compared to mothers. Furthermore, we aimed to investigate if demographic characteristics such as participant gender (male/female), parental status (parent/non-parent) and occupation (academic/non-academic) would have a significant impact on the strength of parent-academia stereotypes. We conducted a 2 (Item: mother vs father) x2 (Category: academic career vs family) x2 (Participant gender: male vs female) x2 (Participant parental status: parent vs non-parent) x2 (Participant occupation: academic vs non-academic) ANOVA was conducted, with repeated

measures on the first two factors. The DV was participant association rating on the 7-point scale.

**Parent-academia stereotypes.**   In order to investigate parent-academia stereotypes and assess whether they vary depending on participant gender, parental status and occupation, a 2 (Item: mother vs father) x2 (Category: academic career vs family) x2 (Participant gender: male vs female) x2 (Participant parental status: parent vs non-parent) x2 (Participant occupation: academic vs non-academic) ANOVA was conducted, with repeated measures on the first two factors. The findings revealed a significant main effect of item, $F(1, 172) = 34.59$, $p < .001$, $\eta 2p = 0.1$, such that participants reported higher association ratings for mothers than fathers, and a significant main effect of category, $F(1, 172) = 550.96$, $p < .001$, $\eta 2p = 0.66$, such that participants reported higher association ratings for words from the family category than the academia category overall. These main effects were qualified by a significant interaction between item and category, $F(1, 172) = 111.54$, $p < .001$, $\eta 2p = 0.23$.

Planned contrast analyses were conducted to follow-up on the significant interaction between item and category. Participants showed significantly stronger associations of fathers with academia ($M = 3.12$, $SD = 1.37$) than mothers, ($M = 2.52$, $SD = 1.12$), $t(172) = 7.01$, $p < .001$, Cohen's $d = 0.48$. Significant differences were also discovered for the family category, with mothers being significantly more strongly associated with family ($M = 6.21$, $SD = 0.65$) than fathers ($M = 5.18$, $SD = 1.05$), $t(172) = 12.03$, $p < .001$, Cohen's $d = 1.18$. Both mothers and fathers were more strongly associated with family than academia, $t(172) = 43.09$, $p < .001$, Cohen's $d = 4.03$ and $t(172) = 24.56$, $p < .001$, Cohen's $d = 1.69$, respectively. These findings suggest that individuals working in Education express strong parent-academia stereotypes reflected in significant differences in associations between mothers/fathers and academia/family (see Fig 1). Interestingly, associations for fathers appear to be much more spread out across the 1–7 scale spectrum, while associations for mothers are much more polarised, showing much lower association scores for academia and higher association scores for family. This finding suggests that role expectations may be more strictly prescribed for mothers compared to fathers.

These findings are consistent with previous research by Correll, Benard and Paik who reported bias against mothers, reflected in lower ratings for their devotion to work and desire for promotion compared to fathers [13]. This perceived lack of commitment to career could reflect the strong stereotypes about mothers revealed in our study and possibly stems from internalised cultural stereotypes about gender roles.

## Study 2

Having established the existence of parent-academia stereotypes in Study 1, we next aimed to investigate our main research questions as to whether a job candidate's parental status, gender and leave status impact their likelihood to be recommended for hire for an academic post. We also investigated whether the evaluator's parent-academia stereotypes predict different career outcomes in academia for job candidates who are parents compared to non-parents. Importantly, these questions were answered with a sample of academics who are conducting research and teaching within the University context, thus allowing us to establish more precisely how academics as a particular occupational group display such biases. Based on the role congruity theory and the stereotype content model, in Study 2 we tested the following hypotheses:

**Hypothesis 1.** Academics will make biased decisions against mothers applying for a job in academia, reflected in a preference to hire women who are not parents compared to women who are mothers (motherhood penalty). In contrast, academics will show a preference to hire men who are fathers compared to men who are not parents (fatherhood premium).

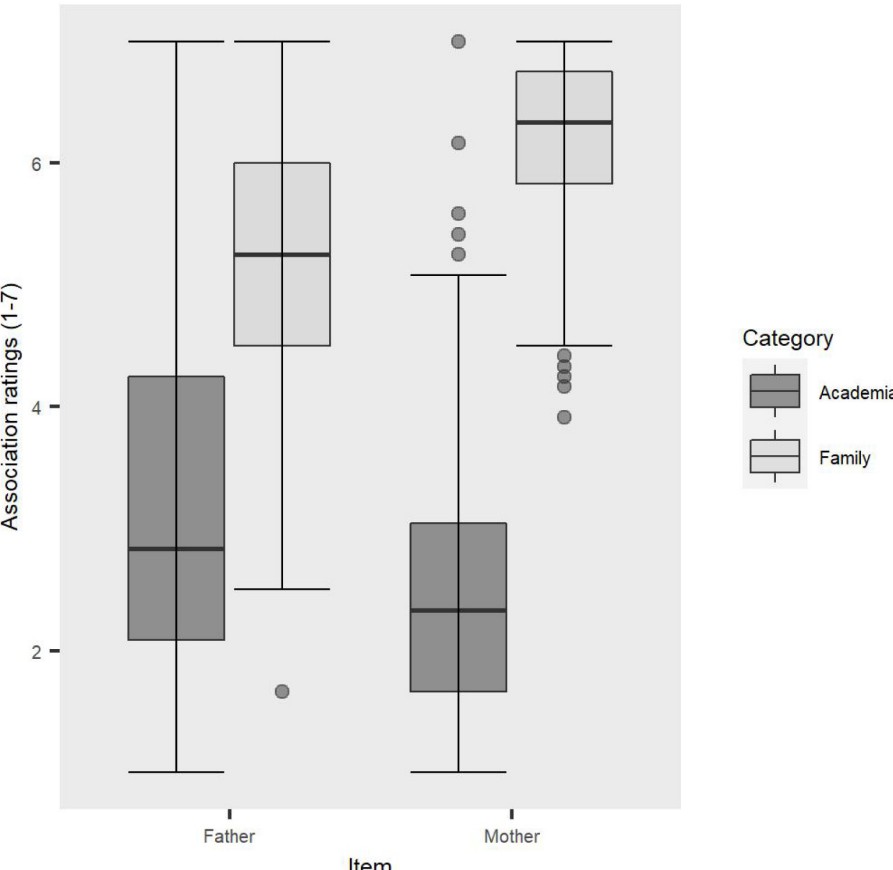

**Fig 1. Significant interaction between item and category, showing differences in participants' associations between mothers/fathers and academia/family.**

**Hypothesis 2.** Bias against leave-takers will be moderated by gender of the candidate in line with the theoretically driven idea that individuals penalise stereotype-inconsistent actions. More specifically, fathers who took paternity leave within a year of applying for the job will be less likely to be recommended for hire than fathers who did not take leave, thus reversing the fatherhood premium effect. In contrast, mothers who did not take maternity leave will be evaluated less favourably than mothers who took leave. This is consistent with role congruity theory, such that individuals who do not fit into the societal expectations will be more likely to be penalised.

**Hypothesis 3.** Individual stereotypes of evaluators will moderate biased decisions, such that academics who have higher parent stereotype scores will make more biased decisions about mothers and fathers in non-traditional roles, such that academics evaluators who display stronger parent-academia stereotypes will be significantly less likely to recommend hiring mothers who did not take maternity leave and fathers who took paternity leave.

Whereas Study 1 attempted to identify what the parent stereotypes in academic and family contexts are, Study 2 aims to assess how these stereotypes predict career evaluations in academia (e.g., hiring for an academic post) and investigate whether biased attitudes about parents result in biased decisions among a sample of academics.

## Methods–Study 2

**Participants–Study 2.** One hundred and twelve academics currently working in universities in the UK and Ireland were recruited to take part in this study via email. The average work experience in academia of our sample was $M = 12.71$ years, $SD = 9.29$. Nearly half of the participants were Lecturers (equivalent of Assistant Professor in the UK system, n = 52), followed by Professors (n = 21), Senior Lecturers (Associate Professor, n = 18), Post-docs/Research fellows/Research associates (n = 12), Readers (Advanced Associated Professor, n = 2), and Teaching Fellows (n = 2). Five participants did not provide information about their post in academia. Our sample consisted of 51 men and 58 women; 3 participants failed to indicate their gender. Fifty-nine participants reported being parents (including stepparents and adoptive parents) while 49 reported not being parents; 4 participants did not indicate their parental status.

**Design and manipulation–Study 2.** This experiment had a mixed design, with three factors: Gender of the job candidate (between-subjects: Male or Female), Parental status of the job candidate (within-subjects: Parent or Non-parent), Leave status of the job candidate (between-subjects: Leave and No leave). The DV was Hiring preferences D score (see Materials and Procedure section). Participants were presented with two matched sets of files, each containing a resume, a fact sheet and an HR memo of two job candidates who applied for an Assistant Professor in Psychology post, where one of the job candidates was a parent and the other one a non-parent. These materials allowed us to manipulate the variables of interest and were adapted from previous work [13] to focus on career progression specifically in academia as a work environment. In the context of the current work, the focus on Psychology serves as an example of an academic field where biases could arise. The resume contained information about the candidate's name, education, academic work experience, publications and grants awarded. The fact sheet contained information about the candidate's gender, marital status, parental status, whether they took any significant leave from work or not, their first date of employment, as well as additional filler information which was matched across conditions such as their ethnicity, nationality, technical skills, courses taught, number of publications. The HR memo was a screenshot of an email addressed to the Search Committee of the university the candidate applied to, containing notes that a member of HR took during a short screening interview via phone with the candidate. This document had some extra filler information establishing that the candidate had enough teaching and research experience and conveyed their motivation for the job. It also re-affirmed the candidate's gender, marital and parental status, and whether they took leave from work in the past year.

The files were pre-tested to ensure that the two candidates were equivalent in terms of qualifications, that the files looked realistic and that the variables of interest were prominent in the documents. Ten academics took part in the pre-test and responded on the 7-point scale. A one-sample t-test revealed that academics deemed the files highly realistic ($M = 6.7$, $SD = .68$) with no individual scores lower than 5, $t(9) = 39$, $p < .001$. When asked whether the two candidates seemed equivalent in terms of qualification (where 1 is 'disagree strongly' and 7 is 'agree strongly'), the mean response was close to the highest point on the scale ($M = 6.5$, $SD = .53$), and a one-sample t-test showed that the candidates were perceived to be of equivalent qualification for the post they are applying for, $t(9) = 31.39$, $p < .001$. In regard to the variables of interest, one sample t-tests showed that academics perceived the leave status ($M = 4.2$, $SD = .8$), $t(9) = 16.84$, $p < .001$, parental status of the candidate ($M = 3.8$, $SD = 1.03$), $t(9) = 11.63$, $p < .001$, and gender of the candidate ($M = 3$, $SD = 1.56$), $t(9) = 6.07$, $p < .001$, as prominent in the files. In conclusion, the pre-test suggested that the job candidates were considered to be

equivalent in terms of qualifications, the files were deemed realistic and the variables of interest were easily noticeable in the files.

In the main experiment, each participant saw two separate files containing information about two job candidates of the same gender, both of whom took leave within a year of applying for the job or not (parental leave for parents and sick leave for non-parents), but one of the job candidates was a parent and the other one a non-parent [13]. The presentation of the parent vs. non-parent candidates was counterbalanced, half of the participants saw the file of the candidate who is a parent first and the file for the non-parent candidate second, the other half vice versa. Additionally, to avoid any bias arising from the files' contents, half of the participants saw the parent job candidate matched with the materials from the first file and the non-parent job candidate matched with the materials from the second file, while the other half saw them vice versa.

**Procedure–Study 2.** The current study was approved by the Faculty of Engineering and Physical Sciences Research Ethics Committee at Queen's University Belfast (reference number EPS 19_186). The study was set up on Qualtrics, a web-based tool for survey implementation and online data collection, and participants were told that they would take part in two separate experiments, the first one aiming to assess how people draw inferences and make decisions based on limited information (hiring decision task) and the second one aiming to assess attitudes towards social issues (parent-academia stereotypes task), to avoid them guessing the purpose of the study.

Participants first completed the hiring decision task that involved reading through two files of job candidates who applied for an Assistant Professor post at a university and evaluating their suitability for the role. Participants were told that they were randomly assigned to the 'thin slice' condition where they would only see a limited amount of information about each of the job candidates, with the purpose of investigating how much detailed information is needed in order to make decisions in processes such as hiring and promotion. After viewing the documents in each job candidate file, participants completed a questionnaire, in which their perceptions of the candidate's suitability for the job were recorded, and afterwards they completed the attention check. Participants were then asked to provide demographic information and, finally, they completed the parent-academia stereotypes task.

**Power analysis–Study 2.** A post-hoc statistical analysis was performed on G*Power for a repeated measures ANOVA with between-subjects and within-subjects factors, a sample of 112 and thirty-two groups with an effect size of $f(U) = 0.35$, calculated based on $\eta2p = 0.11$ as in SPSS, alpha = 0.05. The analysis showed that the current test had power = 0.61.

**Measures–Study 2. The hiring measure** [13, 14, 17, 38, 39] consisted of 3 statements enquiring into the candidate's perceived suitability for the job ("I would recommend hiring this person if I were on the hiring committee", "The candidate should be considered further for the role", "The candidate should be eliminated from consideration for hire" (reverse coded)). Responses were given on the 5-point scale, where 1 is "disagree strongly" and 5 is "agree strongly". Difference scores (D scores) reflecting the difference between hiring ratings for parents vs non-parents were calculated, where the mean hiring score for job candidates who were parents was subtracted from the mean hiring score of non-parent job candidates. Higher D scores therefore indicate a greater likelihood to recommend job candidates who are not parents for hire compared to parents.

*Attention check*. Participants completed 3 attention check questions that asked them to recall the candidate's gender, parental status and whether the candidate took any significant leave from work in the past year, to ensure that they paid attention. These were presented after the questionnaire so as not to create bias in evaluations by increasing the likelihood of participants guessing the purpose of the study and adapting their responses accordingly.

*Demographics*. The demographics questionnaire included questions about participants' gender, parental status, country of origin, current academic post and how long they have worked in academia.

*Parent-academia stereotypes task*. This task was a shortened version of the questionnaire used in Study 1, with 24 questions, 12 assessing associations for "mother" and 12 assessing associations for "father" with words from the "academia" category (word list: professor, research, faculty, lecturer, scholarly, salary) and the "family" category (word list: children, infant, kinship, marriage, household, nurture). In order to assess stereotypes, difference scores (D scores) between stereotype-consistent and stereotype-inconsistent associations were calculated [40]. If respondents associate fathers more strongly with academia and mothers with family (stereotype-consistent) compared to mothers with academia and fathers with family (stereotype-inconsistent), they should give higher association ratings on the 7-point scale to displays that are stereotype-consistent. Higher D scores would therefore indicate stronger stereotypical associations.

## Results and discussion–Study 2

**Analytic plan.** Our first two hypotheses focused on differences in hiring endorsements based on the job candidates' gender, parental status and leave status. We aimed to assess whether hiring decisions in academic contexts vary depending on the parental status of the job candidate, such that female academics who are mothers are less likely to be endorsed to be hired compared to a female academic who is not a parent (motherhood penalty). We also aimed to assess whether male academics who are fathers are more likely to be endorsed to be hired compared to a male academic who is not a parent (fatherhood premium). Additionally, we aimed to assess whether leave status has an impact on the likelihood of parents and non-parents to be endorsed to be hired for an academic job. In order to test these hypotheses, we conducted a 2 (Candidate parental status: parent vs non-parent) x 2 (Candidate gender: male vs female) x 2 (Candidate leave status: leave vs no leave) x 2 (Participant gender: male vs female) x 2 (Participant parental status: parent vs non-parent) ANOVA was conducted, with repeated measures on the first factor. The DV was hiring endorsement.

Our last hypothesis focused on whether the parent-academia stereotypes D score significantly predicts the hiring preferences D score and if this is moderated by Candidate gender and Candidate leave status. To test this hypothesis, we conducted a hierarchical multiple regression analysis, in which we introduced the parent-academia stereotypes D score (mean centred), Candidate gender (dummy coded Male = 0 and Female = 1) and Candidate leave status (dummy coded Leave = 0 and No leave = 1) in the first step, then added in a second step the interaction terms obtained by multiplying the mean centred parent-academia stereotypes D score and Candidate gender, the parent-academia stereotypes D score and Candidate leave status, and Candidate gender and Candidate leave status. Then in a third step we added the three-way interaction term obtained by multiplying the mean centred parent-academia stereotypes D score and the Candidate gender and Candidate leave status variables.

The hiring preferences D score reflected the difference between hiring ratings for parents and non-parents and was calculated by subtracting the mean hiring preferences score for parent job candidates from the mean hiring preferences score of non-parent job candidates. Higher D scores therefore indicated a greater endorsement to hire job candidates who were not parents compared to parents (e.g. more parent bias).

**Hypotheses 1 and 2: Parent bias in hiring decisions and the impact of leave status.** In order to assess if there are differences in hiring endorsements based on the job candidates' gender, parental status and leave status, a 2 (Candidate parental status: parent vs non-parent) x 2

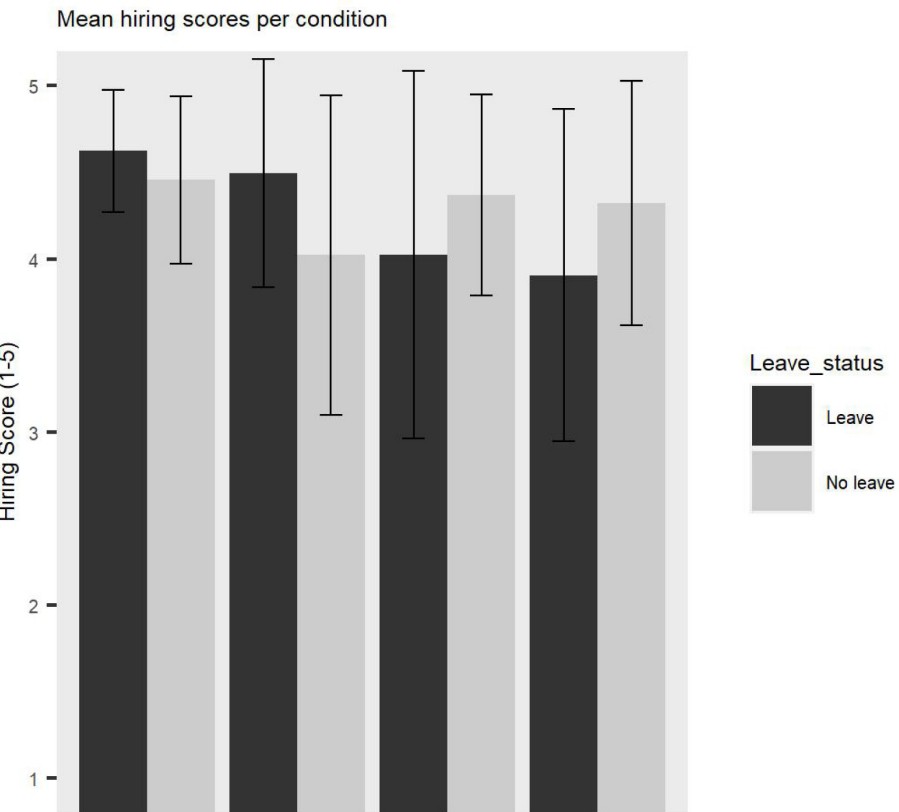

**Fig 2. Descriptive statistics reflecting mean hiring scores.** Error bars represent Standard Deviation (SD).

(Candidate gender: male vs female) x 2 (Candidate leave status: leave vs no leave) x 2 (Participant gender: male vs female) x 2 (Participant parental status: parent vs non-parent) ANOVA was conducted, with repeated measures on the first factor. Fig 2 is a visual representation of mean hiring scores for each condition. The findings revealed a significant main effect of Candidate parental status, $F(1, 91) = 5.79$, $p = .02$, $\eta 2p = 0.1$, such that academics who were not parents were overall significantly more likely to be endorsed to be hired than parents, revealing a significant parent bias in hiring evaluations (see Fig 3). This was not qualified by a significant interaction between Candidate gender and Candidate parental status, $F(1, 91) = 3.54$, $p = .06$.

A significant interaction between Candidate gender and Candidate leave status emerged, $F(1, 91) = 5.87$, $p = .02$, $\eta 2p = 0.1$ (see Fig 4). Planned contrast analyses were conducted to follow up on this interaction and revealed that male job candidates who did not take leave ($M = 4.33$, $SD = 0.57$) received significantly higher hiring scores than male job candidates who did ($M = 4.05$, $SD = 0.88$), $t(91) = 2.19$, $p = .03$, Cohen's $d = 0.38$. In contrast, female job candidates who took leave ($M = 4.55$, $SD = 0.47$) received significantly higher hiring scores than female job candidates who did not take leave ($M = 4.24$, $SD = 0.76$), $t(91) = 2.38$, $p = .02$, Cohen's $d = 0.49$. This finding supports Hypothesis 2 by showing that endorsements for hiring differ significantly based on job candidate gender and leave status, such that job candidates are

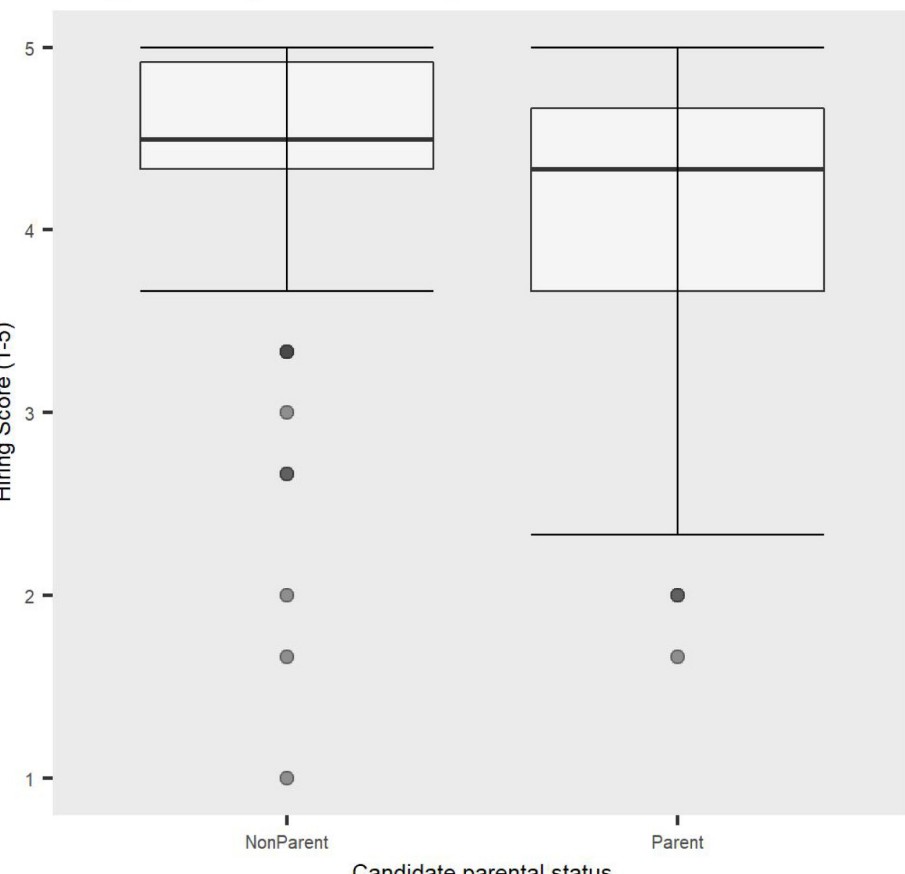

**Fig 3. Parent bias.** Non-parents received significantly higher hiring scores compared to parents. Error bars represent Standard Deviation (SD).

rewarded when they fit into traditional gender roles, such as taking leave due to caregiving reasons for women and not taking leave for men and are penalised when they do not comply with these roles.

Additionally, a significant interaction between Candidate parental status and Participant parental status emerged, $F(1, 91) = 8.51$, $p = .004$, $\eta2p = 0.1$. Planned contrast analyses revealed that academics who were not parents showed an ingroup bias, such that they were significantly more likely to endorse hiring job candidates who were not parents ($M = 4.52$, $SD = 0.45$) compared to candidates who were parents ($M = 4.09$, $SD = 0.91$), $t(91) = 4.55$, $p < .001$, Cohen's $d = 0.60$, while academics who were parents gave similar hiring scores to job candidates who were parents and those who were not parents ($M = 4.27$, $SD = 0.74$ and $M = 4.25$, $SD = 0.78$, respectively), $t(91) = 0.22$, $p = .82$. This finding suggests that the parent bias in academic hiring is driven by evaluators who are not parents (see Fig 5).

A significant four-way interaction between Candidate parental status, Candidate gender, Candidate leave status and Participant parental status was discovered, $F(1, 91) = 11.17$, $p = .001$, $\eta2p = 0.11$. Planned contrast analyses were conducted to follow-up on the significant four-way interaction. The findings showed that women who were mothers ($M = 4.22$, $SD = 0.84$) received significantly lower hiring scores compared to women who were not

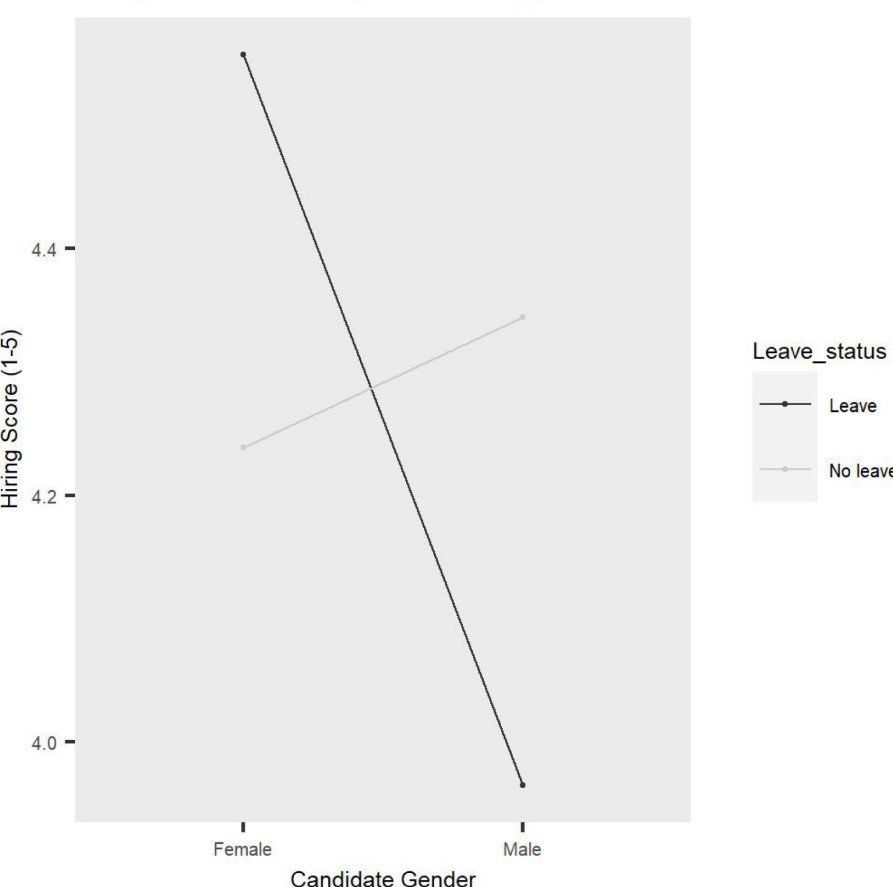

**Fig 4. A significant interaction between candidate gender and candidate leave status.** Hiring ratings vary significantly based on job candidate gender and leave.

parents ($M$ = 4.53, $SD$ = 0.44), $t(91)$ = -5.6, $p$ = .03, Cohen's $d$ = 0.46, providing evidence for the motherhood penalty. However, our findings did not show evidence in support of the fatherhood premium, in that men who were fathers ($M$ = 4.15, $SD$ = 0.85) received similar hiring scores compared to men who were not parents ($M$ = 4.23, $SD$ = 0.81), $t(91)$ = -0.88, $p$ = .38. These findings provide support for the motherhood penalty effect but not for the fatherhood premium.

Overall, these findings reveal parent bias in hiring decisions that impacts both male and female job candidates. Evidence was provided in support of the motherhood penalty, where mothers received significantly lower hiring scores than women who are not parents, but not for the fatherhood premium. In relation to leave status, there were significant differences in how leave-takers were evaluated based on the job candidate's gender–regardless of parental status, men received lower hiring scores and women received higher hiring scores when parents took parental leave and non-parents took sick leave, suggesting that perceptions of gender roles and cultural beliefs about parenthood produce status-based discrimination in the context of academic hiring. Finally, the findings revealed the presence of in-group bias for academics who are not parents when evaluating job candidates for hire but not for academics

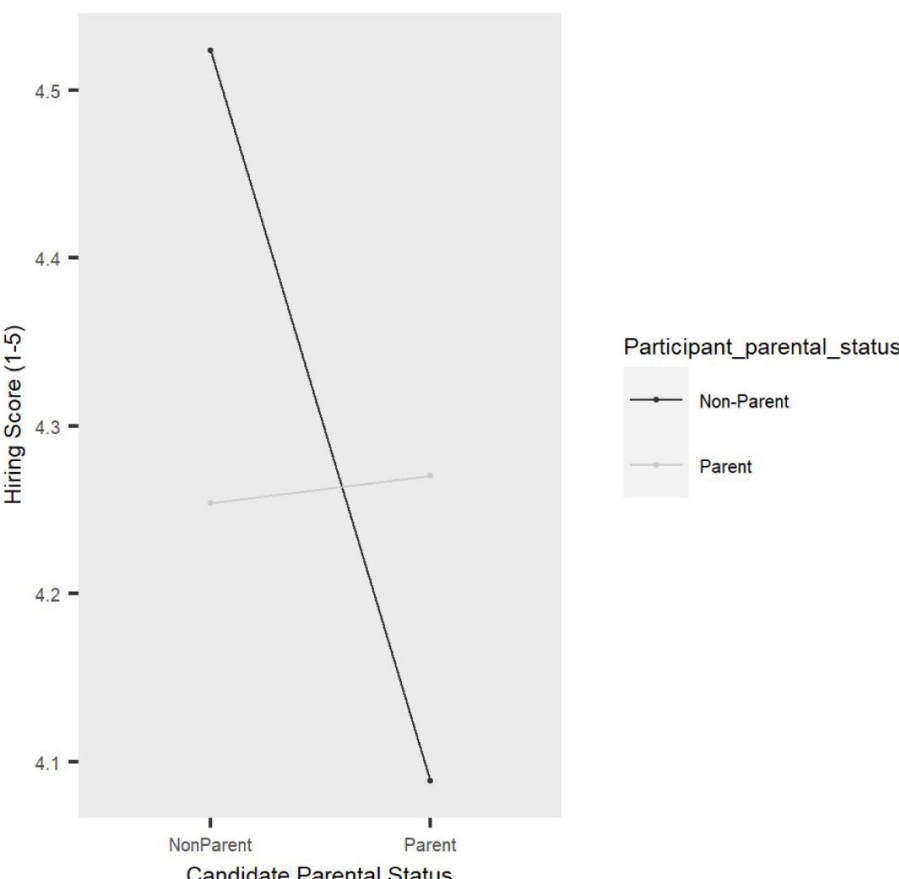

**Fig 5. A significant interaction between candidate parental status and participant parental status showing that the parent bias in hiring is driven by academics who are not parents.**

who are parents, suggesting that academics who are not parents are more likely to discriminate against parents in academic hiring recommendations.

**Hypothesis 3: Parent-academia stereotypes as predictors for hiring.** In order to assess whether the parent-academia stereotypes D score significantly predicts the hiring preferences D score and if this is moderated by Candidate gender and Candidate leave status, we conducted a hierarchical multiple regression analysis. The findings showed that the model explained 8.7% of the variance and predicted hiring preferences D scores significantly, $F(7, 106) = 2.57$, $p = .018$. A significant interaction between parent-academia stereotypes, Candidate gender and Candidate leave status was revealed, $B = 0.63$, $t(106) = 2.31$, $p = .02$. This finding reveals that parent-academia stereotypes significantly predict hiring preferences and that this is moderated by job candidate gender and leave status. To probe this interaction, we plotted and tested the simple slopes for male and female candidates who took or did not take leave (see Fig 6). The slope coefficient for female job candidates who did not take leave was significant, $b = 0.38$, $t(106) = 2.88$, $p = .005$, suggesting that parent bias increased as parent-academia stereotype scores increased. Additionally, the slope coefficient for male job candidates who took leave was also significant, $b = 0.30$, $t(106) = 2.13$, $p = .035$, and the direction of the effect was similar to the one for females who did not take leave, such that parent bias increased as

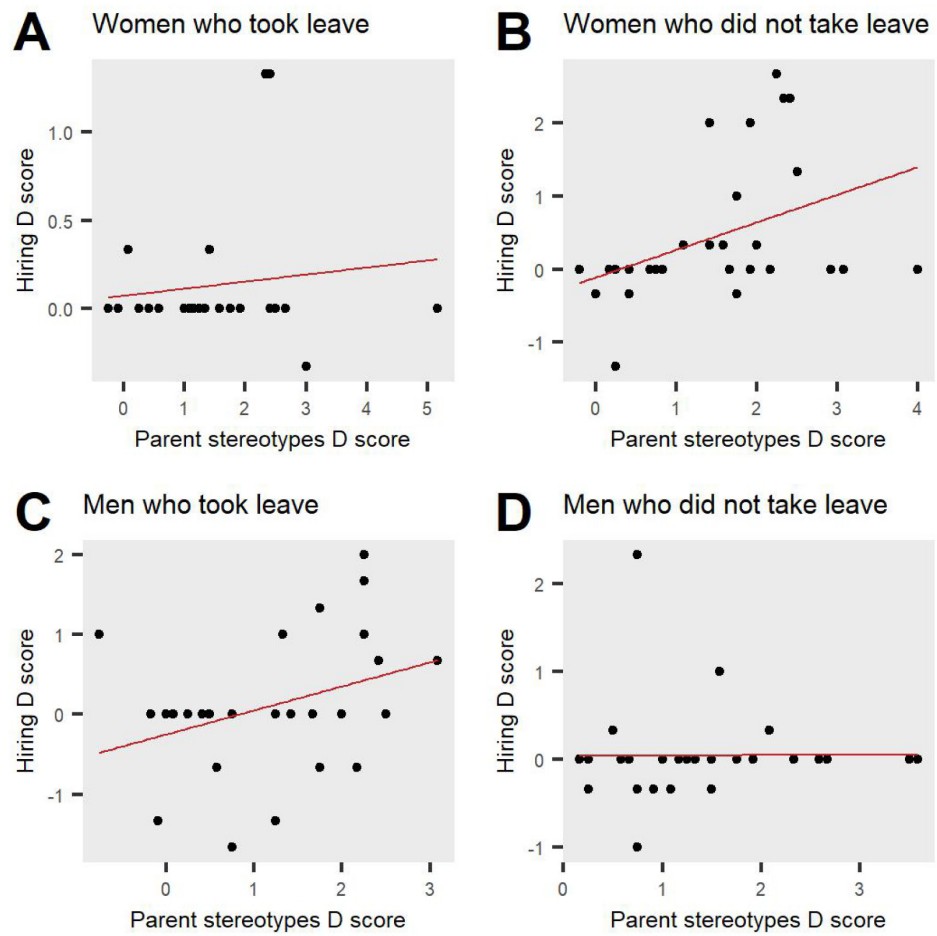

**Fig 6. Parent-academia stereotypes significantly predict biased decisions in hiring.** More stereotypes lead to a reduced likelihood to endorse hiring parents compared to non-parents, specifically men who took leave and women who did not take leave.

parent-academia stereotype scores increased. The slope coefficients for female job candidates who took leave and male job candidates who did not take leave were non-significant, $b = 0.04$, $t(106) = 0.33$, $p = .74$ and $b = 0.01$, $t(106) = 0.035$, $p = .97$, respectively. Overall, these findings revealed that more parent-academia stereotypes predict a significantly reduced likelihood to endorse hiring parents compared to non-parents, specifically when the job candidate is a man who took leave or a woman who did not take leave. This suggests that academics' biased attitudes about parents result in biased behaviours and can significantly impact hiring recommendations, especially in cases where academics evaluate job candidates who have made life choices that are not consistent with cultural sex roles. This finding indicates that role congruity in assessing academics is especially impactful for those who strongly endorse stereotypes, thus suggesting that training to decrease those parent-academia stereotypes may reduce hiring biases against parents.

## General discussion

The goal of the present studies was to identify parent stereotypes in academic and family contexts and assess whether they predict hiring preferences in academia, thus contributing to

explaining the leaky pipeline for female academics. Additionally, we aimed to investigate whether there is hiring bias in academia, such that non-parent job candidates were favoured over parents. The findings provide support for the role congruity theory, showing that individuals in non-traditional roles were penalised in regard to hiring for an academic post and that bias was observed regardless of the gender of the candidate.

Several main findings emerged. First, strong parent-academia stereotypes were revealed that showed significant differences in associations between mothers/fathers and academia/family. Associations of fathers with academia were stronger compared to mothers and of mothers with family compared to fathers, which provides valuable insight into the differences in how men and women are perceived in academic career and family contexts. In addition, there appeared to be more variability in the association scores for fathers and academia/family while mothers received more extreme association scores–higher for family and lower for academia. This could suggest that men are perceived as having more flexibility to operate between the caregiver and breadwinner roles while women are mostly associated with caregiver roles. These findings support and extend previous findings in regard to gender stereotypes, revealing that men are more likely to be associated with work, authority roles and power, while women are linked with family, subordinate roles and communal traits [41].

Second, evidence in support of the motherhood penalty was observed, such that mothers were significantly less likely to be endorsed to be hired than women who are not parents. This finding is consistent with previous research which showed bias against mothers in professional hiring contexts [13, 14]. Interestingly, this was especially the case if mothers did not take maternity leave, which was consistent with the assumption of the role congruity theory that mothers' professional development is hindered by societal expectations for them to be both an ideal worker and an ideal mother. This may create tension when the traditional behaviours associated with these roles are not displayed, such as not taking maternity leave after having a child.

Third, our findings did not provide support for the fatherhood premium but showed that, consistently with the stereotype content model and the role congruity theory, fathers who displayed behaviours that are inconsistent with the cultural stereotype of the breadwinner, such as taking paternity leave, were less likely to be endorsed to be hired than men who did not. This finding suggests that the fatherhood premium effect could be reversed when information about paternity leave taken within a year of applying for an academic job is present in the supporting documentation. This finding is consistent with previous research by Wayne and Cordeiro who reported that male leave-takers received the most negative evaluations compared to female leave-takers and men who did not take leave, showing that parent bias is not restricted to women but could also be experienced by men in situations when caregiving evokes beliefs that the individual is placing caregiving responsibilities before work [26].

Hiring bias was observed for both men and women applying for an academic job in cases when they displayed behaviours that opposed stereotypical cultural norms. Interestingly, the bias was found to be mostly driven by non-parent evaluators, as they were found to display a strong in-group bias and favour the non-parent job candidate compared to the parent job candidate, while no in-group bias was observed for evaluators who were parents. While power in Study 2 of the current work was lower than desirable, the findings provide valuable and novel insight into the processes that may underlie the motherhood penalty effect. These findings highlight an important area for future exploration, focused on the association between caregiver stereotypes and bias against caregivers in career settings.

Finally, circling back to parent-academia stereotypes, we found that the extent to which academic evaluators show bias against mothers and fathers was moderated by evaluators' stereotypes about parents, such that individuals who had more parent-academia stereotypes were

significantly less likely to recommend a parent compared to a non-parent for hire in cases when the job candidate was in a non-traditional role, for example a man who took leave or a woman who did not take leave. This finding showed that evaluators' parent-academia stereotypes are associated with bias against parents in academic hiring contexts, suggesting that academics' endorsement of these stereotypes may significantly impact actual hiring outcomes and, consequently, the career progression of job candidates in non-traditional roles.

## Theoretical implications

The present studies provided theoretical support for the role congruity theory and the stereotype content model within the occupational context of academia, by showing that academic evaluators evaluate mothers and fathers consistent with their gendered expectations–by rewarding those who fit stereotypical expectations and punishing those who do not.

The findings of the current work help to improve current theoretical understanding of the caregiver and breadwinner stereotypes which underlie the motherhood penalty in hiring. The current work showed stronger stereotypic associations of mothers with family compared to academia. This finding is consistent with the caregiver stereotype which implies high warmth and low competence and, according to the stereotype content model, underlies the perceived incongruity between motherhood and traits associated with leadership and results in a motherhood penalty in hiring [14]. In contrast, the current work showed strong associations of fathers with both family and academia, reflected in a greater variability in association scores. This perceived flexibility of fathers in navigating between family and career contexts is consistent with the breadwinner stereotype, which implies high competence and high warmth and could lead to more favourable evaluations in both the workplace and in family contexts. However, the current work did not provide evidence in support of the fatherhood premium, which may have been diminished by the low hiring scores received by fathers who took paternity leave, such that, consistently with previous research, leave-taking behaviour could have signalled caregiver status and caused fathers who took leave to be perceived as conforming to the caregiver stereotype [26].

Moreover, our findings extend these theoretical models and further show that the extent to which individuals are evaluated in terms of their fit depends on evaluators' own parent-academia stereotypes. More specifically, findings showed that individuals who endorsed parent-academia stereotypes more strongly were significantly less likely to recommend a parent for hire compared to a non-parent, specifically when the job candidate is a man who took leave or a woman who did not take leave. This finding suggests that stereotypes about parents can lead to biased hiring endorsements and potentially decisions in a professional context that can directly impact the allocation of organizational rewards. This extends the role congruity theory, showing that the tendency to evaluate academics' suitability for a professional role depends on individual differences, particularly how strongly evaluators endorse associations between parents and academics. To our knowledge, the present study is novel in that it focuses on stereotypes specific to parents as opposed to more general gender stereotypes and shows that individuals' own endorsement of parent-academia stereotypes predicts biased hiring recommendations in academia. Parent-academia stereotypes may also reflect conscious biases related to gender and parenting present in society.

## Practical implications

Apart from providing valuable theoretical insight into the potential impact of parent bias on hiring decisions in academia, the present paper could also generate practical knowledge and policy recommendations about gender issues in the workplace. These recommendations are

especially used in the context of current efforts across academia to ensure gender equality via gender equality initiatives in universities such as Athena SWAN (UK, Ireland), SAGE (Australia), and SEA Change (USA). As most of these initiatives involve setting actions plans to increase gender equality, our findings could serve to inspire action specifically in relation to leave-taking policies and the alleviation of negative attitudes towards individuals in non-traditional roles. Training could be provided to help hiring panel members become aware of parent-academia stereotypes, to understand what they are and how they are expressed, which could alleviate the effects of parent bias in academic hiring. The findings of our study could also help inform policies in the hiring process, for example to omit the job candidate's parental status and leave status from the application documents and encourage a more standardised interview process that does not reveal the parental status of the candidate.

### Limitations and future directions

While our findings support our main hypotheses, one limitation of the present project is that the fictitious job candidates in the study were early career researchers and the findings in relation to hiring bias may not necessarily extend to researchers who are at a later stage of their career. Whereas this is a commonly used methodology in the field [13, 14], future research could profit from an exploration of hiring and promotion decisions at other junctures in academic career development, such as promotion decisions for professorships–a point where more women are lost within the leaky pipeline. Additionally, these effects could be explored across different academic cultures. For example, depending on the culture, academic interviews can be more or less structured, thus leaving room for more bias. As organisational cultures differ in terms of the extent to which they support work-family balance and visibility of family, the organisational contexts and values that allow for parent bias to emerge could be investigated.

Another potential limitation of the current work surrounds possible issues with the measurement of stereotypes. In Studies 1 and 2, parent stereotypes were measured through phrasing the questions as "To what extent do you think people in our society associate the word "mother" with the word "professor"?", in order to avoid social desirability bias. This phrasing of the questions measured participants' cultural stereotypes about parents as they relate to academia and family. Previous research has provided evidence of the overlap between personal and cultural stereotypes [36]. However, it has been argued that cultural stereotypes stem from the social dimension of intergroup beliefs and that there is a distinction between participants' personal beliefs about group characteristics and what they think the social stereotypes about these characteristics are, at least among low-prejudice people [42]. Therefore, the findings of the current work concerning personal stereotypes about parents should be interpreted with caution, as the relationship between personal and cultural stereotypes could be dynamic and variable.

The small sample size and, consequently, low power in Study 2 is another potential limitation of the current work. Low power creates additional difficulty in interpreting the findings, as it reduces the chance for a true effect to be detected and increases error rates. While the current work shows interesting and important findings and a strong indication for an association between parent stereotypes and bias against parents in hiring contexts, the findings of Study 2 should be interpreted with caution due to the small sample size. Future research could profit from attempting to replicate these findings using a similar paradigm but ensuring a larger sample size.

A direction for future research could be to implement a more inclusive approach in the study of the challenges that caregivers face in academia, particularly in relation to gender. The current work focused on binary gender, in line with previous research in the field, and

investigated how male and female caregivers are perceived in academic contexts. However, we acknowledge that the focus on men and women as a dichotomy could be seen as a limitation of the current work, as there are individuals who identify with other gender identities which are outside of this traditional binary. We recognise that there is currently a gap in the field regarding the experiences of caregivers of non-binary gender and how they are perceived in academic and family contexts, and future research could focus on addressing this gap.

In addition, race/ethnicity is also important to consider in the study of parent bias effects in academia. Based on the assumption that bias in professional contexts may be stronger for individuals who are at intersections of multiple marginalised identities [43], mothers from minority ethnic backgrounds could experience even more barriers compared to white mothers. Therefore, future research that investigates challenges for caregivers in academia could focus not just on gender, caregiver status and leave status in the study of parent bias, but also on ethnic background.

Future research could also explore parent bias during the pandemic. Our project was conducted before the pandemic and the pandemic may just have widened the gap, especially for individuals in non-traditional roles, as previous research has shown that crises tend to deepen existing inequalities [44]. Future research could also consider recruiting larger participant samples to investigate parent biases in academic settings, as well as focus on studying potential distinctions in parent bias effects between different disciplines.

## Conclusion

The present paper aimed to provide insight into the nature and expression of parent-academia stereotypes and whether they predict professional ratings and potentially outcomes in academia. Our findings provided evidence for the presence of hiring bias against parents in academic settings and revealed that individuals' own endorsement of stereotypes impact their evaluations of parents in a hiring context in academia. The present study provided novel information about parent stereotypes in academic career and family contexts that offers a basis for further exploration of how these parent-academia stereotypes are expressed and what they predict in terms of career decisions in academia.

## Author Contributions

**Conceptualization:** Vasilena Stefanova, Ioana Latu.

**Data curation:** Vasilena Stefanova.

**Formal analysis:** Vasilena Stefanova.

**Investigation:** Vasilena Stefanova.

**Methodology:** Vasilena Stefanova, Ioana Latu.

**Project administration:** Vasilena Stefanova.

**Resources:** Ioana Latu.

**Supervision:** Ioana Latu.

**Validation:** Vasilena Stefanova.

**Visualization:** Vasilena Stefanova.

**Writing – original draft:** Vasilena Stefanova.

**Writing – review & editing:** Vasilena Stefanova, Ioana Latu.

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
