## [Decision Letter · Decision Letter 0]

16 May 2022

PONE-D-22-00288Navigating the leaky pipeline: Do stereotypes about parents predict career outcomes in academia?PLOS ONE

Dear Dr. Stefanova,

Thank you for submitting your manuscript to PLOS ONE. After careful consideration, we feel that it has merit but does not fully meet PLOS ONE’s publication criteria as it currently stands. Therefore, we invite you to submit a revised version of the manuscript that addresses the points raised during the review process.

 I agree with the reviewers concerns regarding statistical power, lack of detail in the data analytic plan, and methodology. Addressing these concerns are critical for meeting the PLOS ONE's criteria for publication.

We look forward to receiving your revised manuscript.

Kind regards,

Kristina Hood, Ph.D.

Academic Editor

PLOS ONE

Journal Requirements:

2. Please note that in order to use the direct billing option the corresponding author must be affiliated with the chosen institute. Please either amend your manuscript to change the affiliation or corresponding author, or email us at plosone@plos.org with a request to remove this option.

Reviewers' comments:

Reviewer's Responses to Questions

**Comments to the Author**

1. Is the manuscript technically sound, and do the data support the conclusions?

Reviewer #1: Yes

Reviewer #2: Partly

Reviewer #3: Yes

2. Has the statistical analysis been performed appropriately and rigorously? 

Reviewer #1: Yes

Reviewer #2: Yes

Reviewer #3: No

3. Have the authors made all data underlying the findings in their manuscript fully available?

Reviewer #1: Yes

Reviewer #2: Yes

Reviewer #3: Yes

4. Is the manuscript presented in an intelligible fashion and written in standard English?

Reviewer #1: Yes

Reviewer #2: No

Reviewer #3: Yes

5. Review Comments to the Author

Reviewer #1: Introduction. Sentence commencing A gender imbalance in visible…. Does not quitter make sense to me. Perhaps a rewording. I think this first paragraph needs to be strengthened. Why does it matter so much that there might be a gender imbalance? How does this impact several key issues not only at the level of the academic but also at a societal level? Whilst not wishing to step aside from the question of equipoise I can see that not have an adequate and fair representation of females in academic at all levels but certainly at higher levels means we society loses out concerning not maximizing opportunities to promote those who will be highly equipped to address key societal research questions. This is regrettable.

Page 3 line 47 – back up the sentence that mock hiring tasks that are widely used in the field with some evidence to support this statement.

Line 48 understanding or examining? Perhaps examining better

Are you only focusing on female academics working in STEM subjects? Perhaps this needs to be made clear as it may well be (is0 present in the arts and humanities too. Need for justification for this is needed.

Page 4 line 79 – support this statement with evidence that mothers still perform the majority of caregiving duties.

Throughout the introduction and background, I think it would be safer to maintain some degree of equipoise concerning the study question. Perhaps state may or might as opposed to an unquestioning assumption that a leaky pipeline effect could reinforce stereotypes.

Can you please contextualize Correll, Benard and Paik’s study? Where did it take place? And when?

Similarly for Cuddy, Fiske and Glick paper.

Interesting aim set out on page 6.

Emphasis on theory strengthens this paper. And is clearly explained.

Page 8, line 8 – perhaps better to use word hypothesize rather than propose

Page 9, line 190 – I am a little confused here. Has study 1 been completed? I would change predicted to hypothesize.

Set out the aims, objectives of the study along with hypotheses before providing a discrete section that provides a rationale for the study design would be useful before proceeding to explain the two studies. This would be present at the beginning of the Methods section and before Participants.

Explain what a ‘Prolific platform’ is. Why £3? Why was an incentive or inducement necessary? What field of education. Specify, please.

Please provide a section on data analysis and how the various questions for this study were examined. At the moment this is interspersed in the section Results and discussion, the latter which should be separated from results. See below where this is done to much better effect

Page 13 lines 275-281 explain what these tests mean concerning the study findings.

Page 14 lines 299-301, I would reserve discussion/interpretation of study findings to the discussion section. Moreover, no need to reflect on study findings alongside the wider literature here. This is a discussion.

Good to see hypotheses here in study 2.

I think a little thought needs to go into how the paper is presented across the two studies where methods are explained.

Be explicit that participants refer to either study 1 or 2 in the sub-headings.

Design and manipulation for study 2 should read analysis for study 2. A clear exposition of the method of analysis is required concerning the hypotheses suggested earlier on.

Then move to findings from study 2.

Mention of research ethics approval needs to go much earlier in this paper to cover both studies.

Explain what Qualitrics is to the reader.

The results of this study are presented clearly and are fascinating.

Please defer discussion to a separate discussion section of this paper rather than interspersing it into the results.

Overall, you present a very interesting discussion but it is rather bereft of any references where you have examined your data alongside the wider evidence. This is regrettable and lets down an otherwise interesting story. Please go back to the literature to critically examine how your findings are supported by wider evidence or refute it and why this might be the case.

Reviewer #2: Comments to Author(s)

MS# PONE-D-22-00288

"Navigating the leaky pipeline: Do stereotypes about parents predict career outcomes in

academia?”

This study uses innovative experimental methods to inquire about the presence and implications of gender, parent, and parental leave-taking biases among UK adults. It focuses on their implications for hiring endorsements in academia, but first uses some experimental methods for exploratory analyses among 180 UK workers in Education. Then, the study turns to a main analysis of the responses of 112 UK and Irish academics to experimental manipulations. The topic is interesting and important (although that is not a reviewed criteria for this journal) and much of the study is well done. However, much more precision is necessary in referencing the cultural stereotypes indicators and in referencing what the dependent variable indicates. There is also a need for rather substantial copyediting mostly surrounding the use of plural nouns and in verb tense.

1. As described on pp. 11-12, respondents were asked about cultural stereotypes (e.g., to what extent do you think that people in society associate the word…). In my opinion, it is important to refer to this indicator as “perceptions of cultural stereotypes” or “recognitions of cultural stereotypes” or similar—and there is way too much slippage on this in the manuscript.

For example, there is need to edit:

-Title and Short Title(use “perceptions of stereotypes”?)

-2nd to last sen of Abstract (use “perception of more parent-academia stereotypes”

-Lns 176-177 “evaluators’ own perceptions of stereotypes”

-Ln 179: “evaluators recognize stereotypes of those groups”

-Ln 185 “individuals’ own perceptions of stereotypical…”

-Ln 191 “academics who perceive stronger”

-Ln 194 “will be less likely to recommend hiring”

-Ln 251 use participants’ “perceptions of” or “recognitions of” cultural stereotypes

-Ln 308: “perceptions of”

-Ln 327: Perceptions of cultural stereotypes by evaluators

-Ln 330: “who recognize”

-Ln 333: “perceptions of” or “recognitions of”

-Ln 334: “commonly biased”

-Ln 440: “perceive that fathers are more associated”

-Ln 555: “recognize cultural stereotypes”

-Ln 556-557 “suggesting that cultural change is needed and that training to decrease parent-academia stereotypes may reduce hiring biases against parents.”

-Ln 606: “perceived stereotypes” and then “individuals who recognized”

-Ln 609: “evaluators’ perceptions of”

-Ln 619: “own perceptions of”

-Ln 620-621 “who recognized...to a greater extent”

-Ln 627: “recognize stereotypical associations”

-Ln 629: “own perceptions”

-Ln 669: “own recognitions”

2. There is also a need to be more precise with what the DV indicates. As described on pp. 18-19, it indicates perceptions of the suitability of the candidates for being hired or “hiring ratings.”

Thus, there is a need to edit:

-Title (use “endorsements for hiring”?)

- Short Title (use “hiring preferences” or “hiring recommendations”?)

-3rd sen of Abstract (use “endorsements for hiring men versus women” instead of “hiring outcomes”)

-4th sen of Abstract (use “less likely to be endorsed to be hired”)

-5th sen of Abstact (use “biased hiring recommendations” and then “reduced likelihood to endorse hiring”)

-Ln 124: (use “..if there was evidence that parent-academia…”

-Ln 125: “to recommend hiring a parent…”

-Ln 129: “hiring recommendations”

-Ln 307: “recommended for”

-Ln 322: “to be endorsed to be hired”

-Ln 330: “recommend hiring”

-Ln 333: “career evaluations”

-Ln 450-454: “to be endorsed to be hired” 3x

-Ln 455: “hiring endorsements” or “hiring support”

-Ln 460: “an indicator of hiring support” or “hiring endorsements”

-Ln 462-463: “ to be endorsed to be hired”

-Ln 477: “endorsements for hiring”

-Ln 486: “endorse hiring” or “support hiring”

-Ln 516: “academic hiring endorsements”

-Ln 529: “hiring preferences D”

-Ln 530-531 “hiring preferences score” 2x

-Ln 532: “greater endorsement”

-Ln 538: “hiring preferences”

-Ln 549: “to endorse hiring”

-Ln 552: “biased hiring preferences that may significantly impact..”

-Ln 559: “to endorse hiring”

-Ln 563: “hiring preferences”

-Ln 580: “to be endorsed to be hired”

-Ln 589: “to be endorsed to be hired”

-Ln 607: “to endorse hiring”

-Ln 611: “may be significantly impacting actual hiring”

-Ln 621: “ to endorse hiring”

-Ln 624: “biased hiring endorsements and potentially decisions…”

-Ln 630: “predict biased hiring endorsements in academia”

-Ln 667: “professional ratings and potentially…”

-Ln 670: “impact their feelings”

3. In terms of other copyediting notes:

I suggest editing the first sen. of the Abstract so it reads: “The motherhood penalty seemingly reflects a preference to hire and employ female workers who are

not parents compared to mothers, however, little is known about whether this effect

is attributable to parent stereotypes per se.”

4. I suggest deleting the leaky pipeline reference in the title—it is not precise enough.

5. Ln 50: “an academic career”

6. Ln 57: insert cites for above claims

7. Ln 58/59/62/106—use plural “biases”

8. Ln 61: end sen. after “record.” Start next w/ “They also rated..”

9. Ln 66-68: “…may increase the likelihood that women leave academia, contributing to the leaky pipeline phenomenon.”

10. Ln 76: “It has therefore been suggested that mothers experience heightened barriers in the workplace that go beyond those associated with gender per se.”

11. Ln 99: use plural “contexts”

12. Ln 106: “has focused”

13. Ln 117: use “impacts” and then delete last sen. of par. and first sen of next par. (lns 118-122).

14. Ln 123: After deletions, start sen. with: “Overall, then, the aim of…”

15. Ln 127: “…about corresponding social groups.”

16. Ln 128: “delete the larger picture in gender and parent bias by assessing”

17. Ln 134: “requirements of members of..”

18. Ln 135-136 “discrepancies often arise due to the gendered…”

19. Ln 137: “expected to be ideal workers…”

20. Ln 149: careers, plural. The next sen. needs to be deleted or clarified (Lns 149-153).

21. Ln 190-191: predict (present tense) and “will impact”

22. Ln 197: “as is commonly”

23. Ln 221-222: Delete redundant first part of sen. and begin with: “Seventy-six of the respondents”

24. Ln 254-257: After edits regarding this issue, can delete this sen. Or, you may want to rephrase and clarify that “We consider perceptions of cultural beliefs to proxy personal beliefs, also.”

25. Lns 441-442 use plural mothers/fathers

26. Ln 504: delete “strong”

27. Ln 511: “… when parents took parental leave and non-parents took sick leave”

28. Ln 519: “the hiring D”

29. Ln 520: delete “hierarchical”

30. Ln 579: delete “strong”

31. Ln 630-631: delete “These strong” and end w/ “, too”

32. Ln 634: “the potential impact”

Reviewer #3: Thank you for the opportunity to review this manuscript! Given the continual underrepresentation of women and substantial gender biases that emerge in academia, this work is much needed. This work employed two studies. The first explored the existence of parent-academia stereotypes, finding endorsement of perceptions of parent-academia stereotypes, with fathers more associated with academia than mothers. The second study examined hiring decisions given candidate gender, parental status, and leave status, along with participant parental status and gender. Results provide support for the motherhood penalty; mothers, particularly those who took leave, were less likely to be hired. Interestingly, a bias of non-parent participants emerged, such that non-parents were more likely to hire other non-parents, regardless of gender.

Although this work is incredibly important, and is no doubt novel, major concerns with statistical power, along with some methodological concerns, must be addressed if this work is to be accepted.

Introduction:

1.P. 8 , line 181 – “individuals who displayed more gender stereotypes,” does this refer to the gender stereotypes that one holds and endorses (e.g., one believes women should fit feminine gender roles), or one’s own adherence to gender stereotypes (e.g., one believes that they fit feminine gender roles)? Please clarify.

2.The present work is well-based in role-congruity theory, with good discussion of stereotype content model.

Method:

Study 1

1.Why specifically target people in education, particularly if more than half of them are not academics? Justify

2.Who specifically pre-tested the academia/family stimuli? Grad students? Post-docs? Professors? Also, were there any concerns that academics’ knowledge of research and experimental design might impact their responses?

3.The authors have designed their questions to seemingly reflect perceptions of cultural stereotypes, instead of cultural stereotypes. To what extent do perceptions of cultural norms accurately reflect both cultural and individual stereotypes?

Study 2

1.The authors say the variables of interest were prominent – how was prominence and salience of the variables balanced with obscuring the purpose of the study (ie., not making the variables too obvious)?

2.In addition to the attention check items, did the authors include any validity check questions to assess whether participants guessed the purpose of this study or not?

Results:

Study 1

1.There are some concerns raised about power in this study, given the 2 x 2 x 2 ANOVA, with 180 participants. Were the groups for the ANOVA fairly evenly distributed? Was there sufficient power for the effects found? Also, please present effect sizes for these analyses.

Study 2

1.Similar to Study 1, there are concerns about power here, perhaps more so considering the 2 x 2 x 2 x 2 x 2 ANOVA, with only 112 participants. Given that there are 32 groups in this ANOVA, was there sufficient power for the effects found? Again, please provide effect sizes for these analyses.

Discussion:

1.Generally strong summary of results, though it is difficult to comment on the conclusions drawn given the concerns about power listed above.

2.The authors may consider expanded the implications’ sections to discussion possibilities for future research. For instance, how might the added layer of race change the results found in the present work?

3.The authors may also consider adding expanded discussion of the stereotype content model in the ‘Theoretical Implications’ section and how the present results connect to stereotype content model. It is less clear how results support the SCM and are related to it, given that the studies did not examine warmth and competence directly.

6. PLOS authors have the option to publish the peer review history of their article (what does this mean?). If published, this will include your full peer review and any attached files.

Reviewer #1: **Yes: **Dr Jonathan Koffman

Reviewer #2: No

Reviewer #3: No

---

## [Author Response · Author response to Decision Letter 0]

22 Jun 2022

Dear Dr Hood:

We would like to thank you and the Reviewers for the insightful and helpful reviews of our manuscript “Navigating the leaky pipeline: Do stereotypes about parents predict career outcomes in academia?” (PONE-D-22-00288). We appreciate the constructive guidance toward ways in which we can strengthen this manuscript.

In our revision, we focused on and addressed the points that the Reviewers identified in the Decision Letter. We believe that these changes have greatly improved the manuscript, and we hope that you agree with this assessment.

In what follows, we discuss how we addressed all concerns brought up by the Reviewers. We state each point that was raised in the Decision Letter and the reviews, and we follow each comment with our response and changes. For clarity we numbered the reviewer comments and, where appropriate, we grouped similar comments together. 

Again, we thank you and the reviewers for the very helpful suggestions concerning the previous manuscript. We believe that the revised manuscript is substantially improved, and we hope that you will agree that the paper would be of interest to readers of Plos ONE. This manuscript is not currently under review elsewhere and has not been previously published either in whole or in part. All research participants were treated in accordance with APA ethical standards. 

Best regards, 

The authors

Reviewer 1, Point 1. Introduction. Sentence commencing A gender imbalance in visible…. Does not quitter make sense to me. Perhaps a rewording. I think this first paragraph needs to be strengthened. Why does it matter so much that there might be a gender imbalance? How does this impact several key issues not only at the level of the academic but also at a societal level? Whilst not wishing to step aside from the question of equipoise I can see that not have an adequate and fair representation of females in academic at all levels but certainly at higher levels means we society loses out concerning not maximizing opportunities to promote those who will be highly equipped to address key societal research questions. This is regrettable.

We thank Reviewer 1 for bringing our attention to this issue and for their helpful suggestions. We have followed Reviewer 1’s suggestions and have strengthened our rationale for studying gender imbalances in the first paragraph, page 3 lines 44-51. This section reads:

“Failure to recruit and retain female faculty through academic promotion leads to gender imbalances in university leadership [3]. The lack of women in senior academic positions shows a severe underrepresentation of women in academic decision making, as members of committees and recruitment panels. This gender imbalance in scientific leadership results in an underuse of the expertise and skills of a significant part of the Higher Education workforce [12]. It is therefore essential to address gender inequalities in academia, in order to ensure a diversity of perspectives in scientific leadership.”

Reviewer 1, Point 2. Page 3 line 47 – back up the sentence that mock hiring tasks that are widely used in the field with some evidence to support this statement.

We thank Reviewer 1 for bringing this to our attention. We have included references to previous research by Correll, Benard and Paik (2007) and Cuddy, Fiske and Glick (2004) who have used mock hiring tasks on page 3 lines 52-55, page 9 lines 197-199 and page 31 line 721, respectively. Those sections read:

“Using the theoretical framework of role congruity theory [8] and experimental methods employing mock hiring tasks that are widely used in the field [13, 14], the present paper focuses on examining these gender biases in academia, especially as they interact with biases about parents”

“After investigating the content of parent-academia stereotypes in Study 1, we proceed to test the above predictions using an experimental method involving a mock hiring decision, as is commonly used in the previous literature [18, 22].”

“Whereas this is a commonly used methodology in the field [18, 22], future research could profit from an exploration of hiring and promotion decisions at other junctures in academic career development, such as promotion decisions for professorships – a point where more women are lost within the leaky pipeline.”

Reviewer 1, Point 3. Line 48 understanding or examining? Perhaps examining better

We thank Reviewer 1 for their suggestion. We have replaced “understanding” with “examining” on page 3 line 54.

Reviewer 1, Point 4. Are you only focusing on female academics working in STEM subjects? Perhaps this needs to be made clear as it may well be (is0 present in the arts and humanities too. Need for justification for this is needed.

We thank Reviewer 1 for raising this valid point. As mentioned in the Methods sections of our paper, in Study 2 of our paper, the fictitious job candidate was a psychologist applying for an Assistant Professor in Psychology post and the participants who evaluated the candidate were academics from a variety of fields. In Study 1 we recruited individuals working in Education which included academics from different disciplines, as well as other areas of Education such as college and university administration, secondary school teaching, etc. Therefore, the focus of the paper is not particularly on STEM but we take Psychology as an example field where such biases can arise. We have clarified this on page 17 lines 384-386:

“In the context of the current work, the focus on Psychology serves as an example of an academic field where biases could arise.”

We have also mentioned the distinction between different disciplines in terms of parent bias effects as a potential direction for future research on page 33, lines 763-765:

“Future research could also consider recruiting larger participant samples to investigate parent biases in academic settings, as well as focus on studying potential distinctions in parent bias effects between different disciplines.”

Reviewer 1, Point 5. Page 4 line 79 – support this statement with evidence that mothers still perform the majority of caregiving duties.

We thank Reviewer 1 for bringing this to our attention. We have followed Reviewer 1’s suggestion and have included references to previous research that provided evidence that mothers still perform the majority of caregiving duties. Please see this on page 5 lines 84-85.

Reviewer 1, Point 6. Throughout the introduction and background, I think it would be safer to maintain some degree of equipoise concerning the study question. Perhaps state may or might as opposed to an unquestioning assumption that a leaky pipeline effect could reinforce stereotypes.

We thank Reviewer 1 for raising this valid point. We have followed Reviewer 1’s suggestion and have replaced assumptions regarding stereotypes with “may” or “might”. Please see this on page 3 line 43, page 4 line 73, page 5 lines 85 and 87, page 6 line 114 and 121, page 7 lines 143, 145, 146 and 149, page 8 lines 173, 174 and 178, page 9 line 181 and 187.

Reviewer 1, Point 7. Can you please contextualize Correll, Benard and Paik’s study? Where did it take place? And when?

Reviewer 1, Point 8. Similarly for Cuddy, Fiske and Glick paper.

We thank Reviewer 1 for their suggestion. We have followed Reviewer 1’s suggestion and have contextualised Correll, Benard and Paik’s and Cuddy, Fiske and Glick’s studies. We have included further clarifications regarding these two papers on page 5 lines 91-92 and page 5 line 98-99, respectively:

“In a study by Correll, Benard and Paik which was conducted in the US in 2007, mothers applying for a marketing position were significantly less likely to be recommended for hire and were offered a significantly lower starting salary than equally qualified childless women [13].”

“These effects were supported by a study by Cuddy, Fiske and Glick, conducted in the US in 2004, which showed that consultants who were mothers were significantly less likely to be hired and recommended for promotion or job-related training compared to childless women [14].”

Reviewer 1, Point 9. Interesting aim set out on page 6.

Reviewer 1, Point 10. Emphasis on theory strengthens this paper. And is clearly explained.

Reviewer 1, Point 18. Good to see hypotheses here in study 2.

Reviewer 1, Point 25. The results of this study are presented clearly and are fascinating.

We thank Reviewer 1 for their positive feedback on our manuscript and appreciate the enthusiasm that they have expressed for our research.

Reviewer 1, Point 11. Page 8, line 8 – perhaps better to use word hypothesize rather than propose

Reviewer 1, Point 12. Page 9, line 190 – I am a little confused here. Has study 1 been completed? I would change predicted to hypothesize.

We thank Reviewer 1 for their suggestion. We have replaced “propose” and “predict” with “hypothesise” on page 8 line 177 and page 9 line 192.

Reviewer 1, Point 13. Set out the aims, objectives of the study along with hypotheses before providing a discrete section that provides a rationale for the study design would be useful before proceeding to explain the two studies. This would be present at the beginning of the Methods section and before Participants.

We thank Reviewer 1 for their suggestion. In our paper we have followed APA guidelines and put the rationale, aims and objectives of the current work at the end of the Introduction. The rationale, aims and objectives presented at the end of the Introduction serve to set the stage for both Study 1 and Study 2, to avoid repetition, as well as to illustrate how the two studies relate to each other and build upon previous research and theory. Please see this section on pages 7, 8 and 9.

Reviewer 1, Point 14. Explain what a ‘Prolific platform’ is. Why £3? Why was an incentive or inducement necessary? What field of education. Specify, please.

We thank Reviewer 1 for raising this valid point. We have clarified what the Prolific platform is on page 10 lines 219-220. This section now reads:

“One hundred and eighty participants working in the field of Education took part in the study via Prolific, an online crowdsourcing platform for participant recruitment, and received £3 for their participation, as per Prolific’s recommendation”

£3 was the participant payment recommended by Prolific to run the study on their online platform. This incentive was necessary, as participant payment is a requirement for data collection on Prolific. Regarding the field of Education, please see an overview of our participants’ professional background on page 10 line 227-231:

“Seventy-six of the respondents reported being professionally involved in academic research or research and teaching combined, while 104 had non-academic jobs, such as college and university administration, IT support, secondary school or college teaching, student unions and campaigns, disability support, etc.”

Reviewer 1, Point 15. Please provide a section on data analysis and how the various questions for this study were examined. At the moment this is interspersed in the section Results and discussion, the latter which should be separated from results. See below where this is done to much better effect

Reviewer 1, Point 21. Design and manipulation for study 2 should read analysis for study 2. A clear exposition of the method of analysis is required concerning the hypotheses suggested earlier on.

Reviewer 1, Point 22. Then move to findings from study 2.

We thank Reviewer 1 for bringing this to our attention. We have followed Reviewer 1’s recommendation and have included an Analytic plan section for Study 1 (page 13 line 283-294) and Study 2 (page 21 line 480-509), respectively, where we have provided an overview of the analyses that we completed in each study to address our hypotheses before we present the findings. These sections read:

“Our first goal was to investigate the content of parent-academia stereotypes by assessing the extent to which mothers are more strongly associated with family than academia compared to fathers and fathers more with academia than family compared to mothers. Furthermore, we aimed to investigate if demographic characteristics such as participant gender (male/female), parental status (parent/non-parent) and occupation (academic/non-academic) would have a significant impact on the strength of parent-academia stereotypes. We conducted a 2 (Item: mother vs father) x2 (Category: academic career vs family) x2 (Participant gender: male vs female) x2 (Participant parental status: parent vs non-parent) x2 (Participant occupation: academic vs non-academic) ANOVA was conducted, with repeated measures on the first two factors. The DV was participant association rating on the 7-point scale.”

“Our first two hypotheses focused on differences in hiring endorsements based on the job candidates’ gender, parental status and leave status. We aimed to assess whether hiring decisions in academic contexts vary depending on the parental status of the job candidate, such that female academics who are mothers are less likely to be endorsed to be hired compared to a female academic who is not a parent (motherhood penalty). We also aimed to assess whether male academics who are fathers are more likely to be endorsed to be hired compared to a male academic who is not a parent (fatherhood premium). Additionally, we aimed to assess whether leave status has an impact on the likelihood of parents and non-parents to be endorsed to be hired for an academic job. In order to test these hypotheses, we conducted a 2 (Candidate parental status: parent vs non-parent) x 2 (Candidate gender: male vs female) x 2 (Candidate leave status: leave vs no leave) x 2 (Participant gender: male vs female) x 2 (Participant parental status: parent vs non-parent) ANOVA was conducted, with repeated measures on the first factor. The DV was hiring endorsement.

Our last hypothesis focused on whether the parent-academia stereotypes D score significantly predicts the hiring D score and if this is moderated by Candidate gender and Candidate leave status. To test this hypothesis, we conducted a hierarchical multiple regression analysis, in which we introduced the parent-academia stereotypes D score (mean centred), Candidate gender (dummy coded Male = 0 and Female = 1) and Candidate leave status (dummy coded Leave = 0 and No leave = 1) in the first step, then added in a second step the interaction terms obtained by multiplying the mean centred parent-academia stereotypes D score and Candidate gender, the parent-academia stereotypes D score and Candidate leave status, and Candidate gender and Candidate leave status. Then in a third step we added the three-way interaction term obtained by multiplying the mean centred parent-academia stereotypes D score and the Candidate gender and Candidate leave status variables.

The hiring preferences D score reflected the difference between hiring ratings for parents and non-parents and was calculated by subtracting the mean hiring preferences score for parent job candidates from the mean hiring preferences score of non-parent job candidates. Higher D scores therefore indicated a greater endorsement to hire job candidates who were not parents compared to parents (e.g. more parent bias).”

Reviewer 1, Point 16. Page 13 lines 275-281 explain what these tests mean concerning the study findings.

We thank Reviewer 1 for their suggestion. Following their recommendation, we have clarified the findings of the tests on page 14 line 300-305. This section now reads:

“The findings revealed a significant main effect of item, F(1, 172) = 34.59, p < .001, η2p = 0.10, such that participants reported higher association ratings for mothers than fathers, and a significant main effect of category, F(1, 172) = 550.96, p < .001, η2p = 0.66, such that participants reported higher association ratings for words from the family category than the academia category overall.”

Reviewer 1, Point 17. Page 14 lines 299-301, I would reserve discussion/interpretation of study findings to the discussion section. Moreover, no need to reflect on study findings alongside the wider literature here. This is a discussion.

Reviewer 1, Point 26. Please defer discussion to a separate discussion section of this paper rather than interspersing it into the results.

We thank Reviewer 1 for bringing this to our attention. We welcome the opportunity to clarify our choice of structure for the paper. We have decided to present an initial discussion of our results in the Results and Discussion section of each study and a more in-depth discussion in the overall Discussion section later on. This is a typical structure for a multi-study paper because it saves space and helps avoid repetition and we believe that it serves the purposes of our manuscript well and presents the results and discussion clearly. We hope that the Reviewer and Editor agree this is an appropriate and clear way to structure the paper. 

Reviewer 1, Point 19. I think a little thought needs to go into how the paper is presented across the two studies where methods are explained.

Reviewer 1, Point 20. Be explicit that participants refer to either study 1 or 2 in the sub-headings.

We thank Reviewer 1 for their suggestion. Following this recommendation, we have included clarification in the subheadings for sections that belong to Study 1 or Study 2 on pages 10, 11, 13, 16, 17, 19, 20 and 21.

Reviewer 1, Point 23. Mention of research ethics approval needs to go much earlier in this paper to cover both studies.

We thank Reviewer 1 for bringing this to our attention. Following Reviewer 1’s recommendation, we have provided information about research ethics approval in the Procedure section of Study 1 and the Procedure section of Study 2 on page 11 lines 233-236 and page 19 lines 424-425, respectively:

“The current study was approved by the Faculty of Engineering and Physical Sciences Research Ethics Committee at Queen’s University Belfast (reference number EPS 18_178).”

“The current study was approved by the Faculty of Engineering and Physical Sciences Research Ethics Committee at Queen’s University Belfast (reference number EPS 19_186).”

Reviewer 1, Point 24. Explain what Qualitrics is to the reader.

We thank Reviewer 1 for their suggestion. Following this recommendation, we have clarified what Qualtrics is to the reader in the Measures and Procedure section of Study 1 on page 11 lines 235-236. This section now reads:

“The survey was implemented on Qualtrics, a web-based tool for survey implementation and online data collection.”

Reviewer 1, Point 27. Overall, you present a very interesting discussion but it is rather bereft of any references where you have examined your data alongside the wider evidence. This is regrettable and lets down an otherwise interesting story. Please go back to the literature to critically examine how your findings are supported by wider evidence or refute it and why this might be the case.

We thank Reviewer 1 for raising this point. Following this recommendation, we have examined how our findings relate to previous literature in more depth in the Discussion section, page 27 lines 624-630, page 28 lines 642-647, page 29 lines 669-684:

“These findings support and extend previous findings in regard to gender stereotypes, revealing that men are more likely to be associated with work, authority roles and power, while women are linked with family, subordinate roles and communal traits [41].”

“Second, evidence in support of the motherhood penalty was observed, such that mothers were significantly less likely to be endorsed to be hired than women who are not parents. This finding is consistent with previous research which showed bias against mothers in professional hiring contexts [13, 14].”

“This finding is consistent with previous research by Wayne and Cordeiro who reported that male leave-takers received the most negative evaluations compared to female leave-takers and men who did not take leave, showing that parent bias is not restricted to women but could also be experienced by men in situations when caregiving evokes beliefs that the individual is placing caregiving responsibilities before work [26].”

“The findings of the current work help to improve current theoretical understanding of the caregiver and breadwinner stereotypes which underlie the motherhood penalty in hiring. The current work showed stronger stereotypic associations of mothers with family compared to academia. This finding is consistent with the caregiver stereotype which implies high warmth and low competence and, according to the stereotype content model, underlies the perceived incongruity between motherhood and traits associated with leadership and results in a motherhood penalty in hiring [14]. In contrast, the current work showed strong associations of fathers with both family and academia, reflected in a greater variability in association scores. This perceived flexibility of fathers in navigating between family and career contexts is consistent with the breadwinner stereotype, which implies high competence and high warmth and could lead to more favourable evaluations in both the workplace and in family contexts. However, the current work did not provide evidence in support of the fatherhood premium, which may have been diminished by the low hiring scores received by fathers who took paternity leave, such that, consistently with previous research, leave-taking behaviour could have signalled caregiver status and caused fathers who took leave to be perceived as conforming to the caregiver stereotype [26].”

Reviewer 2, Point 1. As described on pp. 11-12, respondents were asked about cultural stereotypes (e.g., to what extent do you think that people in society associate the word…). In my opinion, it is important to refer to this indicator as “perceptions of cultural stereotypes” or “recognitions of cultural stereotypes” or similar—and there is way too much slippage on this in the manuscript.

For example, there is need to edit:

-Title and Short Title(use “perceptions of stereotypes”?)

-2nd to last sen of Abstract (use “perception of more parent-academia stereotypes”

-Lns 176-177 “evaluators’ own perceptions of stereotypes”

-Ln 179: “evaluators recognize stereotypes of those groups”

-Ln 185 “individuals’ own perceptions of stereotypical…”

-Ln 191 “academics who perceive stronger”

-Ln 194 “will be less likely to recommend hiring”

-Ln 251 use participants’ “perceptions of” or “recognitions of” cultural stereotypes

-Ln 308: “perceptions of”

-Ln 327: Perceptions of cultural stereotypes by evaluators

-Ln 330: “who recognize”

-Ln 333: “perceptions of” or “recognitions of”

-Ln 334: “commonly biased”

-Ln 440: “perceive that fathers are more associated”

-Ln 555: “recognize cultural stereotypes”

-Ln 556-557 “suggesting that cultural change is needed and that training to decrease parent-academia stereotypes may reduce hiring biases against parents.”

-Ln 606: “perceived stereotypes” and then “individuals who recognized”

-Ln 609: “evaluators’ perceptions of”

-Ln 619: “own perceptions of”

-Ln 620-621 “who recognized...to a greater extent”

-Ln 627: “recognize stereotypical associations”

-Ln 629: “own perceptions”

-Ln 669: “own recognitions”

Reviewer 3, Point 5. The authors have designed their questions to seemingly reflect perceptions of cultural stereotypes, instead of cultural stereotypes. To what extent do perceptions of cultural norms accurately reflect both cultural and individual stereotypes?

We thank Reviewers 2 and 3 for raising this valid point and for their suggestions. We acknowledge that the fact that our research did not distinguish between a person’s own stereotypes and cultural stereotypes could be seen as a limitation. However, we argue that there is a significant overlap between perceptions of cultural stereotypes and one’s own stereotypes, based on previous research reporting that there is a significant interrater agreement for personal beliefs and cultural stereotypes (Krueger, 1996; Marks & Miller, 1987). In fact, it has been argued that it may be impossible to measure cultural stereotypes as separate from personal stereotypes (Krueger, 1996). Therefore, in the current research we consider perceptions of cultural beliefs to proxy personal beliefs and have framed our predictions and the discussion of our findings in accordance with this. We have strengthened our rationale for this on page 12 lines 261-270:

“This decision is also supported by evidence of overlap between personal beliefs and cultural stereotypes of other groups [36] and consistent with the stereotype content model which assesses stereotypes from a shared cultural perspective [32]. The social projection model implies that social perceivers tend to estimate prevalent cultural stereotypes based on their own personal beliefs, in that they tend to assume their own beliefs are more common in the general population [37]. Thus, projection results in a positive correlation between a person’s beliefs about the characteristics of a particular social group and their perceptions of the cultural stereotype about the social group [36]. Therefore, in our research we consider perceptions of cultural beliefs to proxy personal beliefs.”

We also recognise that the relationship between personal and cultural stereotypes could be dynamic and variable and have therefore expanded upon this in our Limitations section on page 32, lines 730-742:

“Another potential limitation of the current work surrounds possible issues with the measurement of stereotypes. In Studies 1 and 2, parent stereotypes were measured through phrasing the questions as “To what extent do you think people in our society associate the word “mother” with the word “professor”?”, in order to avoid social desirability bias. This phrasing of the questions measured participants’ cultural stereotypes about parents as they relate to academia and family. Previous research has provided evidence of the overlap between personal and cultural stereotypes [36]. However, it has been argued that cultural stereotypes stem from the social dimension of intergroup beliefs and that there is a distinction between participants' personal beliefs about group characteristics and what they think the social stereotypes about these characteristics are, at least among low-prejudice people [42]. Therefore, the findings of the current work concerning personal stereotypes about parents should be interpreted with caution, as the relationship between personal and cultural stereotypes could be dynamic and variable.”

We have incorporated the suggested change as follows:

- Line 196: replaced “will be less likely to hire” with “would be less likely to recommend hiring”

Reviewer 2, Point 2. There is also a need to be more precise with what the DV indicates. As described on pp. 18-19, it indicates perceptions of the suitability of the candidates for being hired or “hiring ratings.”

Thus, there is a need to edit:

-Title (use “endorsements for hiring”?)

- Short Title (use “hiring preferences” or “hiring recommendations”?)

-3rd sen of Abstract (use “endorsements for hiring men versus women” instead of “hiring outcomes”)

-4th sen of Abstract (use “less likely to be endorsed to be hired”)

-5th sen of Abstact (use “biased hiring recommendations” and then “reduced likelihood to endorse hiring”)

-Ln 124: (use “..if there was evidence that parent-academia…”

-Ln 125: “to recommend hiring a parent…”

-Ln 129: “hiring recommendations”

-Ln 307: “recommended for”

-Ln 322: “to be endorsed to be hired”

-Ln 330: “recommend hiring”

-Ln 333: “career evaluations”

-Ln 450-454: “to be endorsed to be hired” 3x

-Ln 455: “hiring endorsements” or “hiring support”

-Ln 460: “an indicator of hiring support” or “hiring endorsements”

-Ln 462-463: “ to be endorsed to be hired”

-Ln 477: “endorsements for hiring”

-Ln 486: “endorse hiring” or “support hiring”

-Ln 516: “academic hiring endorsements”

-Ln 529: “hiring preferences D”

-Ln 530-531 “hiring preferences score” 2x

-Ln 532: “greater endorsement”

-Ln 538: “hiring preferences”

-Ln 549: “to endorse hiring”

-Ln 552: “biased hiring preferences that may significantly impact..”

-Ln 559: “to endorse hiring”

-Ln 563: “hiring preferences”

-Ln 580: “to be endorsed to be hired”

-Ln 589: “to be endorsed to be hired”

-Ln 607: “to endorse hiring”

-Ln 611: “may be significantly impacting actual hiring”

-Ln 621: “ to endorse hiring”

-Ln 624: “biased hiring endorsements and potentially decisions…”

-Ln 630: “predict biased hiring endorsements in academia”

-Ln 667: “professional ratings and potentially…”

-Ln 670: “impact their feelings”

We thank Reviewer 2 for bringing this to our attention and for their helpful suggestions. We have followed Reviewer 2’s recommendations and have edited the suggested lines where we have referred to the DV in the paper. We have edited the sentences that Reviewer 2 highlighted, and made the following changes:

- Line 1: Short Title: replaced “hiring decisions” with “hiring recommendations”

- Line 27: replaced “hiring outcomes” with “endorsements for hiring men versus women”

- Line 30: replaced “less likely to be hired” with “less likely to be endorsed to be hired”

- Line 32-33: replaced “biased hiring decisions” with “biased hiring recommendations” and “reduced likelihood to hire” with “reduced likelihood to endorse hiring”)

- Line 125: replaced “if parent-academia stereotypes predict” with “if there was evidence that parent-academia stereotypes predict”

- Line 127: replaced “to hire a parent” with “to recommend hiring a parent”

- Line 131: replaced “hiring decisions” with “hiring recommendations”

- Line 332: replaced “their likelihood to be hired” with “their likelihood to be recommended for hire”

- Line 347: replaced “less likely to be hired” with “less likely to be recommended for hire”

- Line 355: replaced “less likely to hire” with “less likely to recommend hiring”

- Line 359: replaced “career outcomes” with “career evaluations”

- Line 484-489: replaced “more likely to be hired” with “more likely to be endorsed to be hired”

- Line 481: replaced “hiring decisions” with “hiring endorsements”

- Line 493: replaced “the DV was hiring decision” with “the DV was hiring endorsement”

- Line 489: replaced “to be hired” with “to be endorsed to be hired”

- Line 535: replaced “hiring decision” with “endorsements for hiring”

- Line 533: replaced “to hire” with “to endorse hiring”

- Line 575: replaced “academic hiring decisions” with “academic hiring recommendations”

- Line 505: replaced “hiring D score” with “hiring preferences D score”

- Line 506-507 replaced “hiring score” with “hiring preferences score”

- Line 508: replaced “greater likelihood to hire” with “greater endorsement to hire”

- Line 584: replaced “hiring outcomes” with “hiring preferences”

- Line 595: replaced “likelihood to hire” with “likelihood to endorse hiring”

- Line 598: replaced “hiring outcomes” with “hiring recommendations”

- Line 604: replaced “likelihood to hire” with “likelihood to endorse hiring”

- Line 609: replaced “hiring outcomes” with “hiring preferences”

- Line 519, Line 629 and Line 640: replaced “to be hired” with “to be endorsed to be hired”

- Line 457, Line 658 and Line 689: replaced “to hire” with “to recommend for hire”

- Line 662: replaced “can significantly impact hiring” with “may significantly impact actual hiring”

- Line 692: replaced “biased behaviours and decisions” with “biased hiring endorsements and potentially decisions”

- Line 699: replaced “predicts biased behaviours in terms of career outcomes in academia” with “predicts biased hiring recommendations in academia”

- Line 769: replaced “professional outcomes in academia” with “professional ratings and potentially outcomes in academia”

- Line 772: replaced “their behaviours towards parents” with “their evaluations of parents”

Reviewer 2, Point 3. In terms of other copyediting notes: I suggest editing the first sen. of the Abstract so it reads: “The motherhood penalty seemingly reflects a preference to hire and employ female workers who are not parents compared to mothers, however, little is known about whether this effect is attributable to parent stereotypes per se.”

-I suggest deleting the leaky pipeline reference in the title—it is not precise enough.

-Ln 50: “an academic career”

-Ln 57: insert cites for above claims

-Ln 58/59/62/106—use plural “biases”

-Ln 61: end sen. after “record.” Start next w/ “They also rated..”

-Ln 66-68: “…may increase the likelihood that women leave academia, contributing to the leaky pipeline phenomenon.”

-Ln 76: “It has therefore been suggested that mothers experience heightened barriers in the workplace that go beyond those associated with gender per se.”

-Ln 99: use plural “contexts”

-Ln 106: “has focused”

-Ln 117: use “impacts” and then delete last sen. of par. and first sen of next par. (lns 118-122).

-Ln 123: After deletions, start sen. with: “Overall, then, the aim of…”

-Ln 127: “…about corresponding social groups.”

-Ln 128: “delete the larger picture in gender and parent bias by assessing”

-Ln 134: “requirements of members of..”

-Ln 135-136 “discrepancies often arise due to the gendered…”

-Ln 137: “expected to be ideal workers…”

-Ln 149: careers, plural. The next sen. needs to be deleted or clarified (Lns 149-153).

-Ln 190-191: predict (present tense) and “will impact”

-Ln 197: “as is commonly”

-Ln 221-222: Delete redundant first part of sen. and begin with: “Seventy-six of the respondents”

-Ln 254-257: After edits regarding this issue, can delete this sen. Or, you may want to rephrase and clarify that “We consider perceptions of cultural beliefs to proxy personal beliefs, also.”

-Lns 441-442 use plural mothers/fathers

-Ln 504: delete “strong”

-Ln 511: “… when parents took parental leave and non-parents took sick leave”

-Ln 519: “the hiring D”

-Ln 520: delete “hierarchical”

-Ln 579: delete “strong”

-Ln 630-631: delete “These strong” and end w/ “, too”

-Ln 634: “the potential impact”

We thank Reviewer 2 for their helpful copyediting suggestions. We have followed Reviewer 2’s suggestions and have made the recommended changes:

- Line 22-24: we have changed the first sentence of the Abstract to “The motherhood penalty seemingly reflects a preference to hire female professionals who are not parents compared to mothers, however, little is known about whether this effect is attributable to parent stereotypes per se.”

- Line 56: replaced “the academic career” with “an academic career”

- Line 63: included references to previous research by Howe-Walsh & Turnbull (2014) and Etzkowitz & Ranga (2011)

- Line 64, 65, 68 and 113: replaced “bias” with “biases”

- Line 67: ended sentence after “record” and started new sentence with “They also rated”

- Line 73-74: replaced “may contribute to the leaky pipeline phenomenon by raising the possibility of women to make the choice to leave academia” with “may increase the likelihood that women leave academia, contributing to the leaky pipeline phenomenon”

- Line 82-83: replaced “It has therefore been suggested that mothers could experience barriers in the workplace in addition to those associated with gender” with “It has therefore been suggested that mothers experience heightened barriers in the workplace that go beyond those associated with gender per se”

- Line 106: replaced “academic context” with “academic contexts”

- Line 113: replaced “focused” with “has focused”

- Line 124: replaced “impact” with “impacts”. Deleted the last sentence of the paragraph and the first sentence of the next paragraph as per Reviewer 2’s recommendations. Started the paragraph on line 125 with “Overall, the aim of”

- Line 129: replaced “about a certain social group” with “about corresponding social groups”

- Line 125: replaced “address the larger picture in gender and parent bias by assessing” with “assess”

- Line 135: replaced “requitements of that social group” with “requirements of members of that social group”

- Line 137: replaced “discrepancy would arise” with “discrepancies often arise”

- Line 138: replaced “are expected to take on the role of the ideal worker” with “are expected to be ideal workers”

- Line 150: replaced “academic career” with “academic careers”.

- Line 150-155: added clarification, such that the sentence now reads: “Such strong stereotypes mean that those individuals perceive the two categories (“motherhood” and “academic career”) as being more incongruent, thus leading to a greater perceived discrepancy between the two roles (mother and academic), which could in turn lead to more negative evaluations of academics who are also mothers, consistent with the role congruity theory.”

- Line 193-194: replaced “hypothesised” with “hypothesise” and “would impact” with “will impact”

- Line 199: replaced “as commonly used” with “as is commonly used”

- Line 227-231: deleted the redundant part of the sentence, such that the sentence now reads: “Seventy-six of the respondents reported being professionally involved in academic research or research and teaching combined, while 104 had non-academic jobs, such as college and university administration, IT support, secondary school or college teaching, student unions and campaigns, disability support, etc.”

- Line 474-476: replaced “father” and “mother” with “fathers” and “mothers”

- Line 569-570: replaced “when they took parental or sick leave” with “when parents took parental leave and non-parents took sick leave”

- Line 699-701: replaced “These strong parent-academia stereotypes may reflect conscious biases related to gender and parenting present in society” with “Parent-academia stereotypes may also reflect conscious biases related to gender and parenting present in society”

- Line 703: replaced “the impact” with “the potential impact”

We have decided to keep the reference to the leaky pipeline phenomenon in the title, as exploring the contributions of bias against parents to the leaky pipeline phenomenon is the main issue of interest in our research. We have also decided to keep the word “hierarchical” on line 579 to describe the multiple regression analysis that we conducted, because in our analysis we added variables to the model in blocks to investigate the moderating effect of Candidate gender and Candidate leave status (e.g., please see Lautenschlager & Mendoza, 1986, https://doi.org/10.1177/014662168601000202)

Reviewer 3, Point 1. Introduction: P. 8 , line 181 – “individuals who displayed more gender stereotypes,” does this refer to the gender stereotypes that one holds and endorses (e.g., one believes women should fit feminine gender roles), or one’s own adherence to gender stereotypes (e.g., one believes that they fit feminine gender roles)? Please clarify.

We thank Reviewer 3 for bringing this issue to our attention. Following their recommendation, we have clarified that on page 9 lines 182-184 we refer to gender stereotypes that one endorses. We have amended this in the manuscript and this line now reads:

“For example, gender stereotypes were found to underlie the gender salary gap, such that individuals who strongly endorsed gender stereotypes were more likely to allocate a higher salary to male employees compared to female employees [34].”

Reviewer 3, Point 2. The present work is well-based in role-congruity theory, with good discussion of stereotype content model.

We thank Reviewer 3 for their positive feedback on our paper and appreciate their constructive guidance toward ways in which we can improve our manuscript.

Reviewer 3, Point 3. Method: Study 1. Why specifically target people in education, particularly if more than half of them are not academics? Justify

We thank Reviewer 3 for raising this valid point. We welcome the opportunity to clarify why we have recruited participants working in Education in Study 1. Due to the exploratory nature of Study 1, we aimed to explore parent-academia stereotypes more broadly and recruited a sample of individuals working in Education. Academia is a part of Education and we aimed to probe parent-academia stereotypes in a broader population, encompassing the field of Education in general. We probed parent-academia stereotypes in this broader population in Study 1 and then focused on academics in particular in Study 2. We have clarified this in the manuscript on page 10 lines 220-223:

“We recruited a sample of individuals working in Education in Study 1 to explore parent-academia stereotypes more broadly. Academia is a part of Education and we aimed to probe parent-academia stereotypes in a broader population, encompassing the field of Education in general.”

Reviewer 3, Point 4. Who specifically pre-tested the academia/family stimuli? Grad students? Post-docs? Professors? Also, were there any concerns that academics’ knowledge of research and experimental design might impact their responses?

We thank Reviewer 3 for bringing this issue to our attention. We pre-tested the academia/family stimuli with junior academics (post-docs and PhD students), their responses were anonymous and there was minimal information about the study in the pre-test. They were only asked to report the extent to which they associate the items with the academia category or the family category. It was necessary to pre-test these items with a sample of academics in particular, in order to assess associations relevant to people from this specific professional background. We did not assess potential knowledge of research as a high risk, as we were interested in assessing associations that are widely perceived within their field, and as such did not test any particular hypotheses that would be subject to demand characteristics. 

Reviewer 3, Point 6. Study 2. The authors say the variables of interest were prominent – how was prominence and salience of the variables balanced with obscuring the purpose of the study (ie., not making the variables too obvious)?

We thank Reviewer 3 for bringing our attention to this issue, and we agree this is a difficult balance to achieve. In our job candidate files, we included several pieces of matched distractor information that could also be considered sensitive information, such as race, age etc., and we hope these have helped in obscuring the purpose of the study. As such, this information was prominent without drawing attention to the study hypotheses. To aid with this, we also put the attention checks at the end, so that participants’ attention was not directed to the specific variables of interest until they completed the part of the study which involved looking at job candidate files. The participants whose data was analysed all passed the attention checks which showed that they had paid attention to the files and remembered the information presented about the candidates.

Reviewer 3, Point 7. In addition to the attention check items, did the authors include any validity check questions to assess whether participants guessed the purpose of this study or not?

We thank Reviewer 3 for their raising this valid point. We regretfully did not include validity check questions in Study 2, however, we safeguarded against participants guessing the purpose of the study through the use of distractor information in the files, as mentioned above in Point 6. We gave participants the opportunity to provide any comments about the study at the end and did not get any comments implying that participants had guessed the purpose of it. Moreover, in an experimental design the potential effect of such confounding variables should be diminished by the practice of random assignment. We are thus confident that this issue did not substantially influence our results. 

Reviewer 3, Point 8. Results: Study 1. There are some concerns raised about power in this study, given the 2 x 2 x 2 ANOVA, with 180 participants. Were the groups for the ANOVA fairly evenly distributed? Was there sufficient power for the effects found? Also, please present effect sizes for these analyses.

We thank Reviewer 3 for bringing this issue to our attention. We apologise for this omission, and have now added a post-hoc analysis to compute the achieved power for the effects found in Study 1 on page 13 lines 277-281. This Power analysis section reads:

“A post-hoc statistical analysis was performed on G*Power for a repeated measures ANOVA with between-subjects and within-subjects factors, a sample of 180 and thirty-two groups with an effect size of f(U) = 0.55, calculated based on η2p = 0.23 as in SPSS, alpha = 0.05. The analysis showed that the current test had power = 0.83.”

We have also added effect sizes for the analyses in Study 1 on page 14.

Reviewer 3, Point 9. Study 2. Similar to Study 1, there are concerns about power here, perhaps more so considering the 2 x 2 x 2 x 2 x 2 ANOVA, with only 112 participants. Given that there are 32 groups in this ANOVA, was there sufficient power for the effects found? Again, please provide effect sizes for these analyses.

Reviewer 3, Point 10. Discussion: Generally strong summary of results, though it is difficult to comment on the conclusions drawn given the concerns about power listed above.

We thank Reviewer 3 for bringing these issues to our attention. Similar to Study 1 above, we have followed Reviewer 3’s recommendations and have included a post-hoc analysis to compute the achieved power for the effects found in Study 2 on page 20 lines 443-447. This Power analysis section reads:

“A post-hoc statistical analysis was performed on G*Power for a repeated measures ANOVA with between-subjects and within-subjects factors, a sample of 112 and thirty-two groups with an effect size of f(U) = 0.35, calculated based on η2p = 0.11 as in SPSS, alpha = 0.05. The analysis showed that the current test had power = 0.61.”

We have also included effect sizes for the analyses in Study 2 on pages 23 and 24. 

In addition, we have included a recommendation regarding sample size in the Limitations and future directions section of the manuscript on page 33 lines 764-766:

“Future research could also consider recruiting larger participant samples to investigate parent biases in academic settings, as well as focus on studying potential distinctions in parent bias effects between different disciplines.”

Reviewer 3, Point 11. The authors may consider expanded the implications’ sections to discussion possibilities for future research. For instance, how might the added layer of race change the results found in the present work?

We thank Reviewer 3 for their suggestion. We have followed Reviewer 3’s recommendation and have expanded the Discussion section to discuss possibilities for future research, including non-binary gender and race/ethnicity on pages 32-33 lines 744-760:

“A direction for future research could be to implement a more inclusive approach in the study of the challenges that caregivers face in academia, particularly in relation to gender. The current work focused on binary gender, in line with previous research in the field, and investigated how male and female caregivers are perceived in academic contexts. However, we acknowledge that the focus on men and women as a dichotomy could be seen as a limitation of the current work, as there are individuals who identify with other gender identities which are outside of this traditional binary. We recognise that there is currently a gap in the field regarding the experiences of caregivers of non-binary gender and how they are perceived in academic and family contexts, and future research could focus on addressing this gap.

In addition, race/ethnicity is also important to consider in the study of parent bias effects in academia. Based on the assumption that bias in professional contexts may be stronger for individuals who are at intersections of multiple marginalised identities [43], mothers from minority ethnic backgrounds could experience even more barriers compared to white mothers. Therefore, future research that investigates challenges for caregivers in academia could focus not just on gender, caregiver status and leave status in the study of parent bias, but also on ethnic background.”

Reviewer 3, Point 12. The authors may also consider adding expanded discussion of the stereotype content model in the ‘Theoretical Implications’ section and how the present results connect to stereotype content model. It is less clear how results support the SCM and are related to it, given that the studies did not examine warmth and competence directly.

We thank Reviewer 3 for their suggestion. In line with this suggestion, we have expanded the Discussion to include an overview of how the present results fit with the predictions of the stereotype content model on pages 29-30 lines 670-685:

“The findings of the current work help to improve current theoretical understanding of the caregiver and breadwinner stereotypes which underlie the motherhood penalty in hiring. The current work showed stronger stereotypic associations of mothers with family compared to academia. This finding is consistent with the caregiver stereotype which implies high warmth and low competence and, according to the stereotype content model, underlies the perceived incongruity between motherhood and traits associated with leadership and results in a motherhood penalty in hiring [14]. In contrast, the current work showed strong associations of fathers with both family and academia, reflected in a greater variability in association scores. This perceived flexibility of fathers in navigating between family and career contexts is consistent with the breadwinner stereotype, which implies high competence and high warmth and could lead to more favourable evaluations in both the workplace and in family contexts. However, the current work did not provide evidence in support of the fatherhood premium, which may have been diminished by the low hiring scores received by fathers who took paternity leave, such that, consistently with previous research, leave-taking behaviour could have signalled caregiver status and caused fathers who took leave to be perceived as conforming to the caregiver stereotype [26].”

---

## [Decision Letter · Decision Letter 1]

18 Aug 2022

PONE-D-22-00288R1Navigating the leaky pipeline: Do stereotypes about parents predict career outcomes in academia?PLOS ONE

Dear Dr. Stefanova,

Thank you for submitting your manuscript to PLOS ONE. After careful consideration, we feel that it has merit but does not fully meet PLOS ONE’s publication criteria as it currently stands. Therefore, we invite you to submit a revised version of the manuscript that addresses the points raised during the review process.I appreciate the changes that were made in your revised submission. However, I agree with Reviewer 3 that the low power of Study 2 needs to be addressed. Please be sure to address Study 2 in your revision.Please ensure that your decision is justified on PLOS ONE’s publication criteria and not, for example, on novelty or perceived impact.

We look forward to receiving your revised manuscript.

Kind regards,

Kristina Hood, Ph.D.

Academic Editor

PLOS ONE

Journal Requirements:

Reviewers' comments:

Reviewer's Responses to Questions

**Comments to the Author**

1. If the authors have adequately addressed your comments raised in a previous round of review and you feel that this manuscript is now acceptable for publication, you may indicate that here to bypass the “Comments to the Author” section, enter your conflict of interest statement in the “Confidential to Editor” section, and submit your "Accept" recommendation.

Reviewer #2: All comments have been addressed

Reviewer #3: (No Response)

2. Is the manuscript technically sound, and do the data support the conclusions?

Reviewer #2: Yes

Reviewer #3: Partly

3. Has the statistical analysis been performed appropriately and rigorously? 

Reviewer #2: Yes

Reviewer #3: Yes

4. Have the authors made all data underlying the findings in their manuscript fully available?

Reviewer #2: Yes

Reviewer #3: Yes

5. Is the manuscript presented in an intelligible fashion and written in standard English?

Reviewer #2: Yes

Reviewer #3: Yes

6. Review Comments to the Author

Reviewer #2: I am satisfied with the authors' edits to the manuscript, in line with the journal's specifications. I commend the authors for their hard work.

Reviewer #3: Thank you for the opportunity to review this manuscript again. It is evident that the authors have taken great care and put much effort into making suggested edits and revisions. However, I still have concerns about the power in Study 2 – it is mentioned, but not fully addressed in the manuscript.

It is appreciated that the authors included power analyses and effect sizes for their studies – this was much needed and helpful for evaluating this manuscript!

For Study 2, power = .61 is considered very low, and there is concern about how the lack of power affects interpretation of results. While it’s a great start to note that perhaps a larger sample size is needed, for this manuscript to be ready for publication, the low power found in Study 2 needs to be addressed head on. The authors might consider including some discussion of how low power might impact interpretation of findings, as well as including the low power in the limitation section in conjunction with discussion of the sample size. I don’t believe the low power to be irredeemable, but acknowledgement of it must be made, and the authors should justify why these results have merit despite this low power.

7. PLOS authors have the option to publish the peer review history of their article (what does this mean?). If published, this will include your full peer review and any attached files.

Reviewer #2: No

Reviewer #3: No

---

## [Author Response · Author response to Decision Letter 1]

24 Aug 2022

Reviewer 3, Point 1. Thank you for the opportunity to review this manuscript again. It is evident that the authors have taken great care and put much effort into making suggested edits and revisions. However, I still have concerns about the power in Study 2 – it is mentioned, but not fully addressed in the manuscript.

It is appreciated that the authors included power analyses and effect sizes for their studies – this was much needed and helpful for evaluating this manuscript!

For Study 2, power = .61 is considered very low, and there is concern about how the lack of power affects interpretation of results. While it’s a great start to note that perhaps a larger sample size is needed, for this manuscript to be ready for publication, the low power found in Study 2 needs to be addressed head on. The authors might consider including some discussion of how low power might impact interpretation of findings, as well as including the low power in the limitation section in conjunction with discussion of the sample size. I don’t believe the low power to be irredeemable, but acknowledgement of it must be made, and the authors should justify why these results have merit despite this low power.

We thank Reviewer 3 for bringing our attention to this issue and for their helpful suggestions. We have followed Reviewer 3’s suggestion and included a paragraph in the limitations section of the General Discussion (page 32 line 747-754) in which we addressed the low power found in Study 2, discussed how low power might impact the interpretation of the findings and included a suggestion for future research to replicate these findings with a larger sample size. We additionally included a brief discussion of the power in Study 2 in the discussion of the findings on page 28 lines 654-657 and a justification of the merit of the findings despite the low power.

“The small sample size and, consequently, low power in Study 2 is another potential limitation of the current work. Low power creates additional difficulty in interpreting the findings, as it reduces the chance for a true effect to be detected and increases error rates. While the current work shows interesting and important findings and a strong indication for an association between parent stereotypes and bias against parents in hiring contexts, the findings of Study 2 should be interpreted with caution due to the small sample size. Future research could profit from attempting to replicate these findings using a similar paradigm but ensuring a larger sample size.”

“While power in Study 2 of the current work was lower than desirable, the findings provide valuable and novel insight into the processes that may underlie the motherhood penalty effect. These findings highlight an important area for future exploration, focused on the association between caregiver stereotypes and bias against caregivers in career settings.”

---

## [Editor Report · Decision Letter 2]

21 Sep 2022

Navigating the leaky pipeline: Do stereotypes about parents predict career outcomes in academia?

PONE-D-22-00288R2

Dear Dr. Stefanova,

We’re pleased to inform you that your manuscript has been judged scientifically suitable for publication and will be formally accepted for publication once it meets all outstanding technical requirements.

Kind regards,

Kristina Hood, Ph.D.

Academic Editor

PLOS ONE
---

## [Editor Report · Acceptance letter]

26 Sep 2022

PONE-D-22-00288R2 

Navigating the leaky pipeline: Do stereotypes about parents predict career outcomes in academia? 

Dear Dr. Stefanova:

I'm pleased to inform you that your manuscript has been deemed suitable for publication in PLOS ONE. Congratulations! Your manuscript is now with our production department. 

Kind regards, 

on behalf of

Dr. Kristina Hood 

Academic Editor

PLOS ONE